Resource

# A single-cell spatial chart of the airway wall reveals proinflammatory cellular ecosystems and their interactions in health and asthma

Régis Joulia [1,9] ✉, Sara Patti[1], William J. Traves[1], Lola Loewenthal[1,2,3], Laura Yates[1], Simone A. Walker[1], Franz Puttur[1], May Al-Sahaf[1], Katherine N. Cahill [4], Juying Lai [5,6], Salman Siddiqui[1,7], Joshua A. Boyce[5,6], Elliot Israel [8] & Clare M. Lloyd [1,9] ✉

Determining spatial location of cells within tissues gives vital insight into the interactions between resident and inflammatory cells and is a critical factor for uncoupling the mechanisms driving disease. Here, we apply single-cell spatial transcriptomics to reveal the airway wall landscape in health and during asthma. We identified proinflammatory cellular ecosystems that exist within discrete spatial niches in healthy and asthma samples. These cellular hubs are characterized by a high level of chemokine and alarmin expression, along with unique combinations of stromal cells. Mechanistically, we demonstrated that receptors, such as ACKR1, retain immune mediators locally, while amphiregulin-expressing mast cells are prominent within these proinflammatory hubs. Despite anti-inflammatory treatments, the asthma airway mucosa exhibited a distinct remodeling program within these cellular ecosystems, marked by increased proximity between key cell types. This study provides an unprecedented view of the topography of the airway wall, revealing distinct, specific ecosystems within spatial niches to target for therapeutic intervention.

The recent development of monoclonal antibodies targeting type 2 inflammation has dramatically improved therapeutic options for individuals with asthma[1,2]. Despite their ability to restrain immune responses, these therapies fail in a significant proportion of patients[3,4]. Further improvement will require a greater understanding of the interactions between infiltrating inflammatory cells and the resident structural cells that constitute the bronchial wall, the primary site of inflammatory responses and tissue remodeling during asthma[5,6].

Recent studies have been transformative for lung biology by redefining the cellular networks during development and disease[7–11]. However, most of these studies rely on enzymatically dissociated tissue and fluidic systems to isolate cells. Consequently, vital information as to the spatial relationships between cell types is lost[12,13].

Here, we use single-cell spatial transcriptomics to interrogate the transcriptional and spatial landscapes of the human bronchial wall during health and asthma. Epithelial and submucosal gland niches

[1]National Heart and Lung Institute, Imperial College London, London, UK. [2]Department of Asthma and Allergy, Royal Brompton and Harefield Hospitals, London, UK. [3]Department of Respiratory Medicine, Royal Brompton and Harefield Hospitals, London, UK. [4]Division of Allergy, Pulmonary and Critical Care Medicine, Vanderbilt University Medical Center, Nashville, TN, USA. [5]Departments of Medicine and Pediatrics, Harvard Medical School, Boston, MA, USA. [6]Jeff and Penny Vinik Center for Allergic Disease Research, Division of Allergy and Clinical Immunology, Brigham and Women's Hospital, Boston, MA, USA. [7]Department of Respiratory Sciences, University of Leicester, Leicester, UK. [8]Division of Pulmonary and Critical Care Medicine, Department of Medicine, Brigham and Women's Hospital, Boston, MA, USA. [9]These authors jointly supervised this work: Régis Joulia, Clare M. Lloyd. ✉e-mail: r.joulia@imperial.ac.uk; c.lloyd@imperial.ac.uk

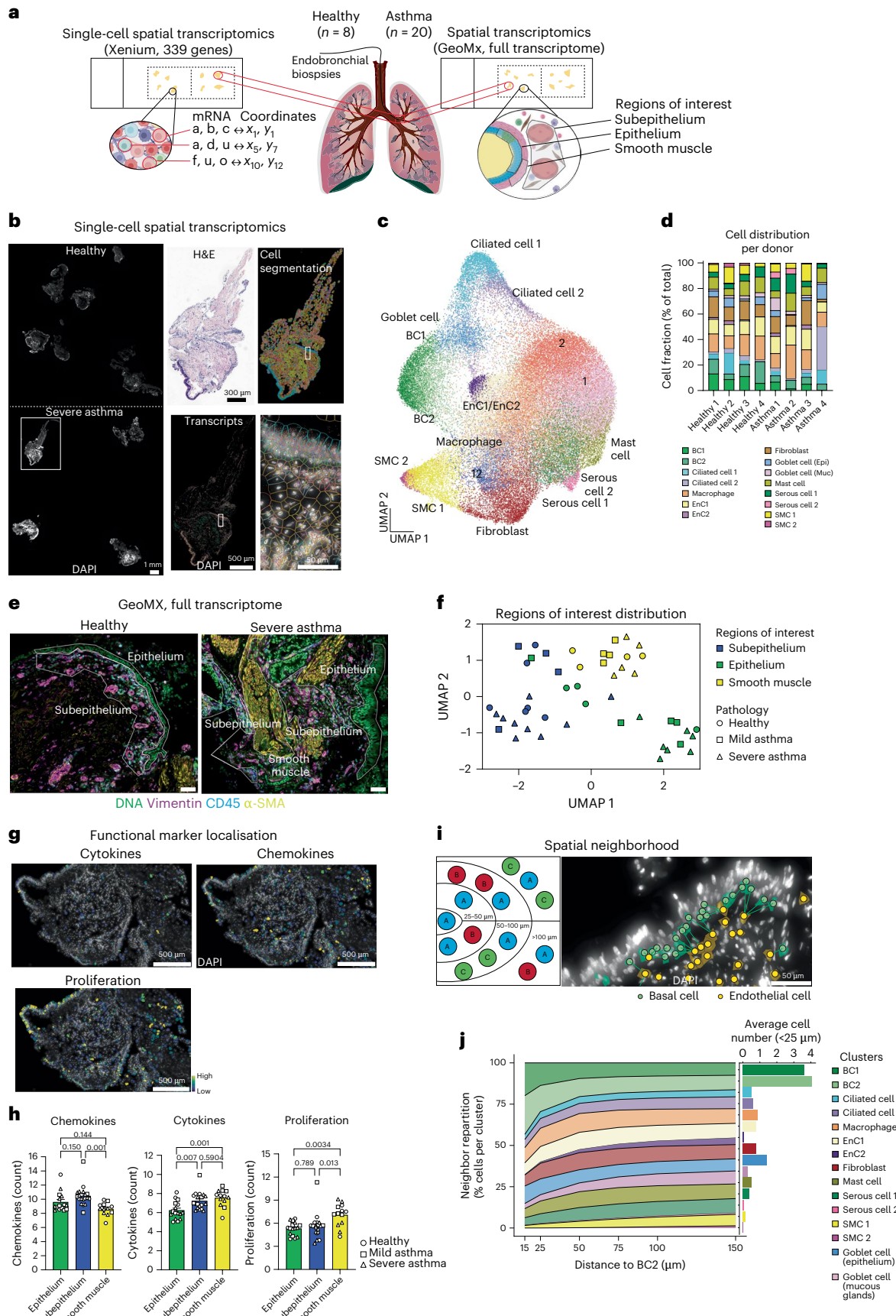

**Fig. 1 | Single-cell spatial transcriptomics to analyze lung wall heterogeneity and cellular interaction. a**, Schematic showing the site of lung endobronchial biopsies and Xenium and GeoMX slide placement. **b**, Overview of tissue sections on a Xenium slide and zoom-in images of one section (white box) with corresponding H&E staining, Xenium cell segmentation, total transcripts detected and superimposed DAPI, cell segmentation and transcripts in one area (data are representative of two independent experiments). **c**, UMAP projection of all 60,567 cells captured using Xenium with the indicated cluster names identified. **d**, Cell fraction per donor (cohort 1; see Supplementary Table 1 and Supplementary Data 1 for participant information). **e**, Images of endobronchial biopsies stained for DNA (Syto83, green), vimentin (purple), CD45 (blue) and α-SMA (yellow) showing the selected ROIs (boxed regions, representative of four independent experiments); scale bars 100 μm. **f**, UMAP of all 43 ROIs according to their localization and pathology ($n = 4$ donors in each group, cohort 3; see Supplemental Table 1 for participant information). **g**, Cytokines, chemokines and proliferation genes shown as a transcript density map. **h**, Chemokines, cytokines and proliferation genes in different ROIs and pathology status ($n = 15$ (epithelium), 19 (subepithelium) and 13 (smooth muscle) ROIs from $n = 4$ donors per group, cohort 3); data are shown as mean ± s.e.m. **i**, Schematic depicting nearest neighbor analysis and a Xenium image showing the nearest endothelial cell (yellow dots) from each basal cell (green dots). Lines represent shortest distance (data are representative of two independent experiments). **j**, Repartition of cells and distance to BC2 for each cluster and average neighboring cells within 50 μm; SMCs, smooth muscle cells; Epi, epithelium; Muc, mucous glands. Data in **h** were analyzed by one-way analysis of variance (ANOVA), followed by a Tukey's post hoc test. Illustrations were created with Adobe Illustrator and Biorender.com.

exhibit discrete ecosystems containing cells that produce alarmins and chemokines. Furthermore, we provide evidence of the molecular regulators of these ecosystems, such as atypical chemokine receptor 1 (*ACKR1*) and mast cells expressing amphiregulin (*AREG*). Finally, in biopsy specimens captured after 24 weeks of exposure to imatinib or matched placebo, we demonstrate the extensive impact of imatinib on the asthma microenvironment and provide a template to analyze drug–target interactions in situ.

## Results

### The spatially resolved chart of the asthmatic airway wall

To determine the spatial transcriptional signature of the bronchial airway, we used the Xenium[14] and GeoMx[15] platforms to analyze endobronchial biopsies from 8 healthy donors and 20 donors with mild to severe asthma exposed to anti-inflammatory treatments (that is, inhaled corticosteroids and biologics; Fig. 1a, Supplementary Table 1 and Supplementary Data 1). With the Xenium platform, we used a 339-gene panel to identify cell lineages and optimized section placements to run healthy and asthma biopsies simultaneously (Fig. 1b, cohort 1, and Supplementary Video 1). We successfully detected more than 2 million transcripts, and unbiased clustering revealed 18 clusters (Fig. 1c). We identified 15 clusters using canonical markers such as Von Willebrand factor (*VWF*) for endothelial cells (Fig. 1c and Extended Data Fig. 1a). We then projected the primary gene signatures (that is, the top five genes expressed) for each cluster onto the lung spatial cell atlas[8] that confirmed our cluster annotations (Extended Data Fig. 1b). It was not possible to discriminate clusters 1, 2 and 12 due to the superimposed expression of genes from multiple different cell types (Extended Data Fig. 1a,b). Of note, all donors contributed to the clusters, irrespective of their pathology status (Extended Data Fig. 1c), and the frequency of each cluster was similar between donors (Fig. 1d). Interestingly, we observed subpopulations of both epithelial and endothelial cells indicating distinct transcriptional and spatial identities (Extended Data Fig. 1d). Additionally, we performed digital spatial profiling in a different cohort of individuals (cohort 3; Supplementary Table 1 and Supplementary Data 1). Tissue sections were stained for DNA, vimentin, CD45 and α-smooth muscle actin (α-SMA) to determine three regions (for example, epithelium, subepithelium and smooth muscle) as previously described[15] (Fig. 1e). We analyzed 48 regions of interests (ROIs)

within these samples, and overall clustering using uniform manifold approximation and projection (UMAP) indicated that ROIs mainly segregated by type of region analyzed (Fig. 1f). Functionally, expression of cytokines, chemokines or proliferation genes was detected in discrete areas in both Xenium (Fig. 1g) and GeoMx data (Fig. 1h). Finally, we explored cellular spatial information to determine the closest neighbor of each cell subset (Fig. 1i). For every cell, we measured its shortest distance to the closest neighbor for a different cluster (for example, every basal cell 2's shortest distance to any endothelial cell). We were able to investigate the neighbor repartition and number of neighboring cells (for example, how many endothelial cells were located around one basal cell) for each cell cluster up to 150 μm (Fig. 1i). For example, basal cells preferentially associated with other basal cells at the closest distance (that is, less than 50-μm range), followed by a close association with goblet cells and, at a longer distance, smooth muscle cells (Fig. 1j).

In summary, we provide a single-cell transcriptional chart of the lung airway wall, map the spatial relationships across 15 clusters of cells in the lungs of healthy individuals and those with asthma and confirm these data using an additional platform and a separate cohort of participants.

### Proinflammatory cellular ecosystems at the airway wall

The airway wall contains distinct regions as shown recently from the spatially resolved general lung cell atlas[8]. In agreement with this, our specific analysis of the airway wall determined that most transcripts were concentrated in two main areas; namely the epithelial–subepithelial area and the mucous gland area separated by a layer of smooth muscle cells (Fig. 2a)[16]. We carefully annotated these two regions in each biopsy section using both hematoxylin and eosin (H&E) and transcript information and avoided any regions with disrupted tissue integrity (that is, dissociated cells from the biopsies; Fig. 2a and Extended Data Fig. 2). Clustering of cells from the epithelial and mucous gland niches from both healthy and asthma samples indicated that our manual selection provided an accurate reflection of the bronchial wall landscape (Fig. 2b,c).

We next analyzed the expression of alarmins (that is thymic stromal lymphopoietin (*TSLP*) and interleukin-33 (IL-33; *IL33*)) and chemokines known to be crucial for lung inflammatory responses during asthma[17,18]. In both niches, we observed discrete areas of high production of these

**Fig. 2 | Lung wall is characterized by discrete proinflammatory hubs. a**, Representative H&E-stained section and matching Xenium image of an asthma biopsy. Transcripts are visualized as a density map and regions used for epithelial and mucous gland niche analysis. All regions are shown in Extended Data Fig. 2. Data are representative of two independent experiments. **b,c**, UMAP of cells captured in epithelial (**b**) and mucous gland (**c**) niches with the indicated names of clusters ($n = 8$ donors, cohort 1). **d**, Representative Xenium image showing inflammatory hubs with transcripts for chemokines (red dots) and alarmins (orange; data are representative of two independent experiments). **e,f**, Gene dot plot for alarmins (that is, *TSLP* and *IL33*) and chemokine expression in stromal cell clusters for epithelial (**e**) and mucous gland (**f**) niches ($n = 4$ donors per group, cohort 1). **g**, Heat map projection of alarmin expression in epithelial and subepithelial ROIs (four to nine ROIs per group from $n = 4$ donors per group, cohort 3). For the gene dot plot, the size of the dot reflects the frequency of cells expressing at least one transcript, and color indicates the level of expression. Columns alternate between healthy (H) and asthma (A) samples. Scales are provided for each plot. Red boxes indicate genes with a $P$ value of <0.05 compared to other clusters, and black boxes indicate genes with a $P$ value of <0.05 between healthy tissue and tissue from individuals with asthma. Data were analyzed by two-sided Mann–Whitney $t$-test (**e** and **f**) and two-way ANOVA, followed by a Dunnett's post hoc test (**g**, healthy vs. mild $P$ value = 0.0002, healthy vs. severe $P$ value = 0.0047).

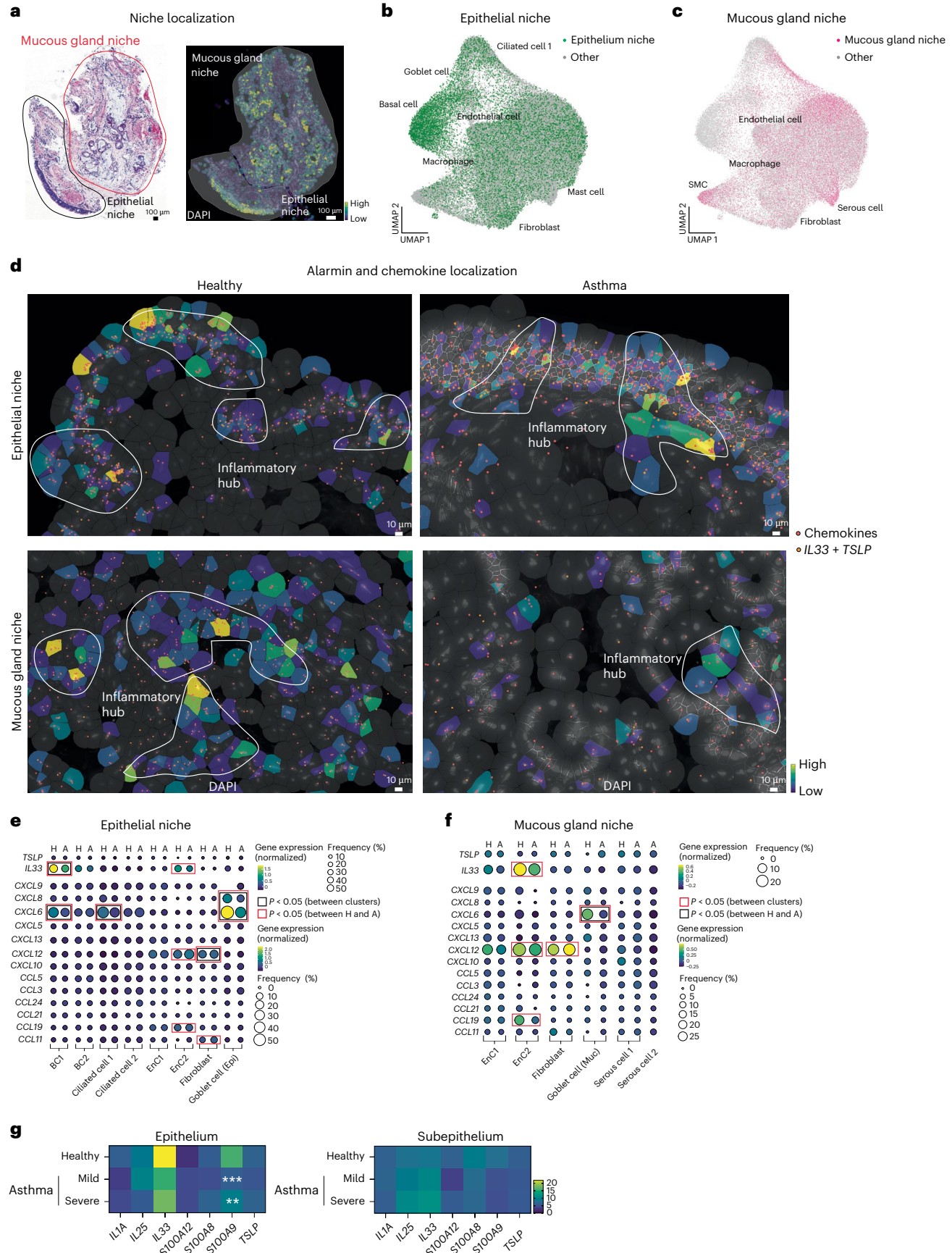

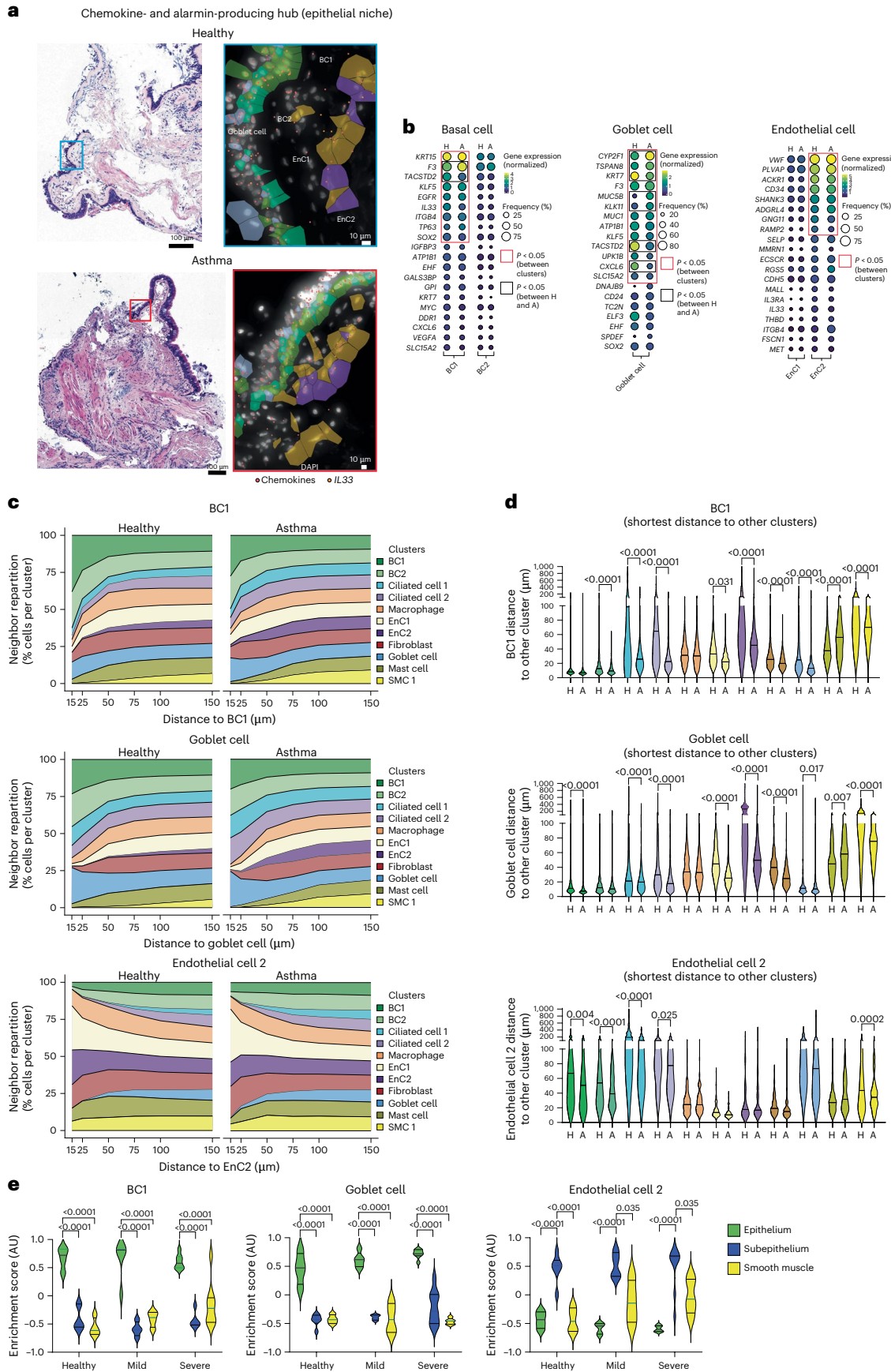

**Fig. 3 | Basal cells, goblet cells and endothelial cells are key regulators of the epithelial niche. a**, Representative H&E image showing the analyzed area (left) and Xenium images of chemokines (right; red dots) and *IL33* (orange) transcript expression by basal cells (light green, BC2; dark green, BC1), goblet cells (light blue) and endothelial cells (dark brown, EnC1; purple, EnC2); data are representative of two independent experiments. **b**, Gene dot plot for the top 20 expressed genes in BC1s/BC2s, goblet cells and EnC1s/EnC2s (*n* = 4 donors per group, cohort 1). **c**, Repartition of neighbors along distance for BC1s (top), goblet cells (middle) or EnC2s (bottom; *n* = 4 donors per group, cohort 1). **d**, Shortest cell-to-cell distance between BC1s (top), goblet cells (middle) or EnC2s (bottom) and other clusters (*n* = 4 donors per group, cohort 1). **e**, BC1, goblet cell and EnC2 enrichment scores (epithelium ROIs *n* = 4 (healthy), 5 (mild) and 7 (severe); subepithelium ROIs *n* = 6 (healthy), 4 (mild) and 5 (severe); smooth muscle ROIs *n* = 4 (healthy), 4 (mild) and 9 (severe) from *n* = 4 donors per group, cohort 3). Columns alternate between healthy (H) and asthma (A) samples. Scales are provided for each plot. Red boxes indicate genes with a *P* value of <0.05 compared to other clusters, and black boxes indicate genes with a *P* value of <0.05 between healthy and asthma samples. Data were analyzed by two-sided Mann–Whitney *t*-test (**b** and **d**) and two-way ANOVA, followed by a Dunnett's post hoc test (**e**); AU, arbitrary units.

mediators (Fig. 2d). At the epithelial side, inflammatory hubs were present just below the outside layer of epithelial cells and present for up to 50 μm inside the tissue. In the mucosal gland, inflammatory hubs were closely associated with cells forming the glands (Fig. 2d). Next, we quantified which cells produce chemokines and alarmins. Stromal cells were among the highest producers of these mediators compared to immune cell clusters (that is, mast cells or macrophages; Fig. 2e,f and Extended Data Fig. 3a). Distinct subpopulations of basal cell 1 (BC1) and endothelial cell 2 (EnC2) were the most positive for *IL33* and chemokines (Fig. 2e). Furthermore, goblet cells exhibited high signal for the neutrophil chemoattractants *CXCL6* and *CXCL8* (Fig. 2d). Within the mucosal gland ecosystem, EnC2s showed the highest expression of *IL33* and chemokines, followed by fibroblasts and goblet cells (Fig. 2f). Individuals with asthma showed a global reduction in alarmin and chemokine expression except *CXCL12*, which perhaps reflects their ongoing treatment regimen that is designed to reduce inflammation (Fig. 2c,d). The former is in line with publicly available single-cell RNA-seq data[11]. We confirmed our observation using the GeoMX platform where we showed that most alarmins were decreased in expression in epithelial and subepithelial areas, most notably *IL33* (Fig. 2g). Similarly, most chemokines followed the same pattern of reduction in epithelial and subepithelial ROIs from the independent cohort (Extended Data Fig. 3b). Interestingly, we noticed a strong increase in *CXCL17* expression in asthma, only at the lung epithelium (Extended Data Fig. 3b). Considering the recently described role for this chemokine in the regulation of T cell migration[19], it is interesting to note that asthma treatment did not impact the expression of this chemokine. Furthermore, genes associated with proliferation were present at low levels in most stromal clusters. Extracellular matrix genes were expressed mainly by fibroblasts and smooth muscle cells but were reduced in asthma (Extended Data Fig. 3c,d).

Collectively, these data clearly indicate that alarmin and chemokines are produced within specific neighborhoods, and asthma treatment reduces the capacity of the cells to produce most chemokines and alarmins.

## Basal, goblet and endothelial cells cooperate at the epithelial niche to produce alarmins and chemokines

We then further characterized alarmin- and chemokine-producing cells within the cellular ecosystems of the epithelial and mucous gland niches (Fig. 3a). BC1s were characterized by increased expression of the genes encoding keratin 15 (*KRT15*; a canonical marker for basal epithelial cells) and coagulation factor III (*F3*; a cell surface glycoprotein important for blood coagulation) compared to BC2s, with expression even more pronounced in asthma biopsies (Fig. 3b). The profile of goblet cells within the epithelium was altered in individuals with asthma compared to in healthy individuals, with increased expression of *CYP2F1* (cytochrome P450, a hemeprotein important for drug metabolism), *MUC5B* (mucin 5B, the main secreted mucin) and *KLK11* and decreased expression of *KRT7* and *TACSTD2* (epithelial glycoprotein; Fig. 3b). In addition, EnC2s were characterized by higher expression of *VWF* (a glycoprotein critical for hemostasis), plasmalemma vesicle-associated protein (*PLVAP*; an endothelial-specific protein involved in the formation of endothelial intracellular microstructures) and, intriguingly, the atypical chemokine receptor 1 (*ACKR1*; a nonsignaling chemokine receptor; Fig. 3b). Of note, we did not detect significant differences in the proportions of these cells between healthy and asthma samples (Extended Data Fig. 4a).

Next, we observed that asthma epithelial niches showed a closer association between these proinflammatory cells (Fig. 3a). Spatial analysis indicated that BC1s, goblet cells and EnC2s were closely associated with themselves, fibroblasts, mast cells and macrophages (Fig. 3c). This set of interactions differed in the bronchial wall of donors with asthma for BC1s and EnC2s, where we observed that these cell types were located much closer to each other and to ciliated cells (Fig. 3c). Interestingly, no differences were observed in the number of neighboring cells for BC1s, goblet cells and EnC2s (Extended Data Fig. 4b). The changes in tissue organization were mainly linked to an increase in cell accumulation. Indeed, we showed that the shortest distance between BC1s, goblet cells and EnC2s and most clusters was reduced in asthma samples (Fig. 3d). This suggests that inflammatory hubs contain more cells in proximity and have an overall increased capacity to generate chemokines and alarmins despite inhaled corticosteroids and biologics. Additionally, we used the gene signature generated with the Xenium platform and cellular deconvolution[20] to confirm the presence of BC1s, goblet cells and EnC2s in our GeoMx data (Fig. 3e). We confirmed our Xenium data by showing that BC1s and EnC2s do exist in an independent cohort, and no differences were observed in cell enrichment between groups (Fig. 3e).

In summary, we identified specific interactions between basal cells, goblet cells and endothelial cells at the airway wall, regulating the formation of proinflammatory interactive hubs. The reduced

**Fig. 4 | Mucous glands show specialized populations of cells with dysregulated activity in asthma. a**, Representative H&E and Xenium images showing the area analyzed and chemokine (red dots) and *IL33* (orange) transcript expression by fibroblasts (dark red) and endothelial cells (light brown, EnC1s; purple, EnC2s). Data are representative of two independent experiments. **b**, Gene dot plot for the top 20 expressed genes in EnC1s/EnC2s (top) and fibroblasts (bottom; *n* = 4 donors per group, cohort 1). **c**, Repartition of neighbors along distance for EnC2s and fibroblasts (*n* = 4 donors per group, cohort 1). **d**, H&E and Xenium images of the area analyzed and goblet cell gene (blue dots), serous cells gene (gray dots), *MUC5B* (red dots) and endothelial cell gene (brown) transcript expression. Data are representative of two independent experiments; scale bar, 100 μm (H&E images). **e**, Gene dot plot of the 20 most expressed genes by goblet cells from mucous glands (*n* = 4 donors per group, cohort 1). **f**, *MUC5B* and *MUC5AC* gene counts (*n* = 9 (healthy), 8 (mild) and 15 (severe) ROIs per group from *n* = 4 donors per group, cohort 3). Data are shown as mean ± s.e.m. **g**, Repartition of neighbors along distance for goblet cells (*n* = 4 donors per group, cohort 1). **h**, Gene dot plot of the 20 most expressed genes by serous cells from mucous glands (*n* = 4 donors per group, cohort 1). **i**, Repartition of neighbors along distance for serous cells (*n* = 4 donors per group, cohort 1). Columns alternate between samples from healthy individuals and those with asthma. Scales are provided for each plot. Red boxes indicate genes with a *P* value of <0.05 compared to other clusters, and black boxes indicate genes with a *P* value of <0.05 between healthy and asthma samples. Data were analyzed by two-sided Mann–Whitney *t*-test (**b**, **e** and **h**) and one-way ANOVA followed by a Tukey's post hoc test (**f**).

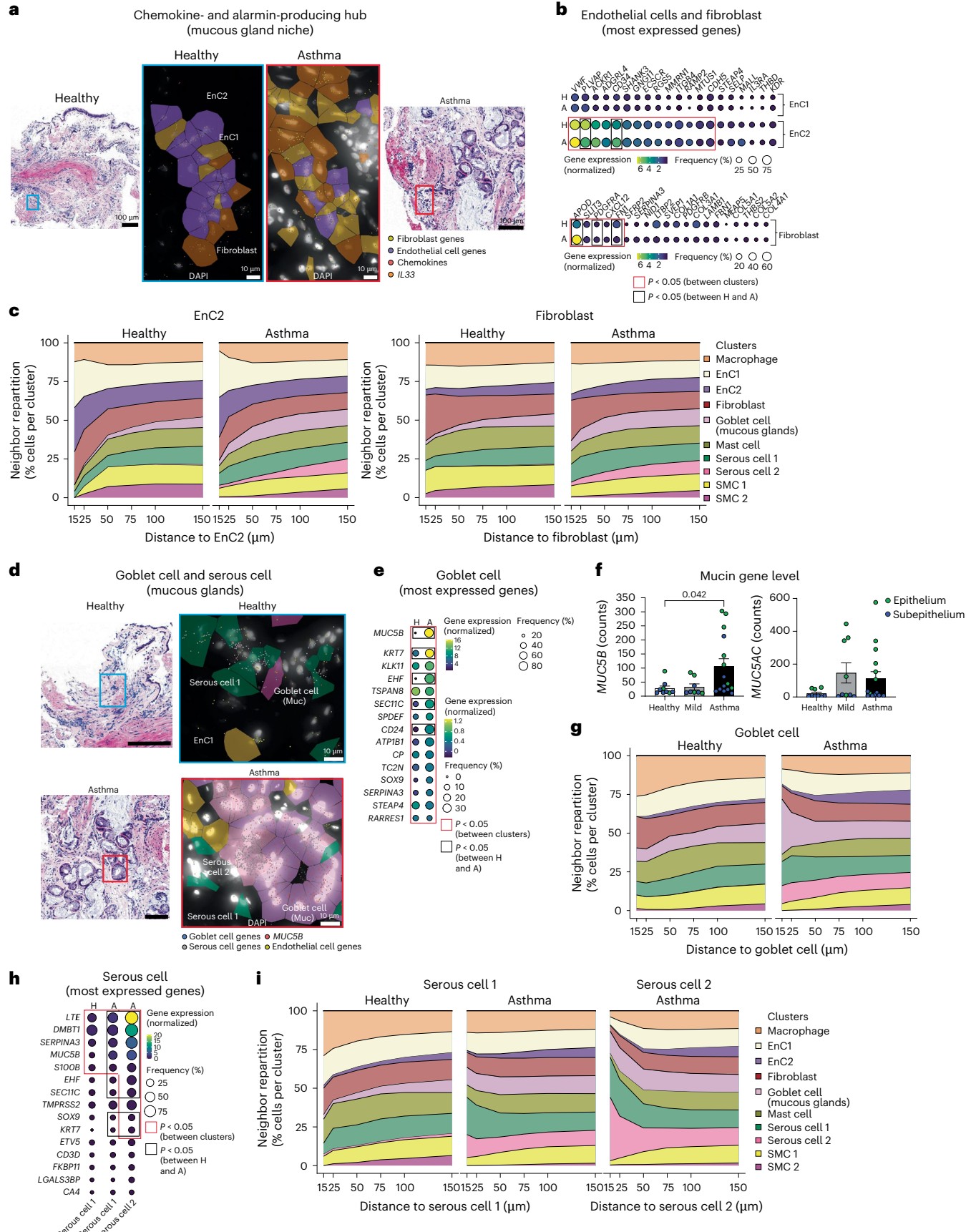

expression of key inflammatory marker genes observed in individuals with asthma is likely driven by anti-inflammatory medication; however, we did observe closer proximity between cell clusters in asthma samples.

## Submucosal gland cell communities are altered in individuals with asthma

We next investigated the distribution of alarmin- and chemokine-producing endothelial cells and fibroblasts within our annotated mucous gland areas (Fig. 4a). No differences were observed in the proportion of all cell clusters in this niche between healthy and asthma tissues (Extended Data Fig. 5a). Endothelial cell clusters showed the greatest changes, with EnC2s exhibiting increased expression of *VWF*, *PLVAP* and *ACKR1* compared to EnC1s (Fig. 4b). Fibroblasts exhibited a relatively homogenous signature between healthy donors and donors with asthma except for the gene encoding apolipoprotein D (*APOD*; a protein that regulates lipid metabolism), which showed ~150% increased expression in fibroblasts from individuals with asthma (Fig. 4b). *APOD* expression was confirmed in our GeoMX cohort where we observed a dose-dependant increase in *APOD* expression according to clinical status (Extended Data Fig. 5b). EnC2s and fibroblasts displayed strong interactions with each other and with the endothelial cell cluster EnC1. This was reinforced in samples from individuals with asthma (Fig. 4c). In parallel, we noted a stronger connection with goblet cells and serous cells (Fig. 4c). Here, again, we did not detect a change in the number of neighboring cells but an increase in the shortest distance between cells, indicating a higher level of clustering in the asthma airway wall (Extended Data Fig. 5c,d). Analysis of goblet cell profiles determined a strong enrichment of *MUC5B* expression, ~340% more in samples from individuals with asthma than in healthy samples (Fig. 4d,e). This was confirmed in our second cohort of participants, with samples from individuals with severe asthma exhibiting higher *MUC5B* expression and a tendency toward increased *MUC5AC* expression (Fig. 4f). Goblet cells were strongly connected with themselves in healthy samples, but also with endothelial cells, fibroblasts and serous cells (Fig. 4g). These close distances were further exacerbated in biopsies from individuals with asthma between goblet cells and serous cells (Fig. 4g and Extended Data Fig. 5e,f).

A key finding here was the identification of a distinct phenotype of serous cells in individuals with asthma. Serous cells are important for the secretion of molecules involved in host defense and epithelium renewal. All serous cells were characterized by the expression of lactotransferrin (*LTF*; a glycoprotein present in secretory fluids with antimicrobial activity) and deleted in malignant brain tumor 1 (*DMBT1*; a conserved protein important for microbial defense)[21]. In the context of asthma, we noted that a distinct serous cell population expressed ~660% more *LTF* than classical serous cells (Fig. 4h). Serous cells existed in close association with most cell types analyzed, notably mast cells, endothelial cells and macrophages (Fig. 4i). In asthma, serous cell 2 cells accumulated in large aggregates but still maintained close interactions with mast cells, macrophages and endothelial cells (Fig. 4i). These changes were due to a shorter distance between cell types but also an increased number of goblet cells and serous cells overall, leading to these aggregates (Extended Data Fig. 5g,h). Serous cells were also present in our GeoMx data, with a potential increase in samples from individuals with asthma (Extended Data Fig. 5i).

Collectively, our data clearly show that the submucosal niche is significantly disrupted in biopsies from individuals with asthma, with aberrant gene expression (for example, *MUC5B*), increased cellular connections between EnC2s, fibroblasts, goblet and serous cells as well as emerging cell types within these proinflammatory ecosystems.

## Mast cells have the capacity to regulate proinflammatory ecosystems

Mast cells and macrophages were the most abundant immune cell types identified at the airway wall. Macrophage signatures within the

epithelial and mucosal niches were comparable between samples from healthy individuals and those with asthma, apart from reduced macrophage receptor with collagenous structure (*MARCO*) expression (Extended Data Fig. 6a) and closer interactions with endothelial cells and serous cells in asthma (Extended Data Fig. 6b,c).

The observed strong interaction between BC1s and EnC2s and mast cells, coupled with our previous work on the role of mast cells in asthma, prompted us to investigate this interaction in more detail[22–31]. Mast cells were abundant within all bronchial biopsies, and imaging data in healthy lung confirmed their localization associated with blood vessels[31,32] (Fig. 5a,b). We obtained a clear transcriptional signature for mast cells using the Xenium platform with the canonical mast cell markers *CPA3* (mast cell protease), *KIT* (stem cell factor receptor) and *MS4A2* (β-chain of the IgE receptor), being easily distinguishable in tissues (Fig. 5c,d). Interestingly, we also found positive signals for effector molecules, such as amphiregulin (*AREG*; a key tissue remodeling cytokine), known to be expressed by mast cells[33]. Although mast cells from donors with asthma exhibited a similar frequency of cells expressing characteristic genes, the level of expression for some markers was lower in those with asthma, notably *CPA3* expression decreased by ~55% (Fig. 5d). Importantly, *AREG* expression was unchanged in asthma samples from epithelial and submucosal niches (Fig. 5d). Analysis of mast cell neighborhood interactions revealed strong connections with EnC1s/EnC2s, macrophages and fibroblasts (Fig. 5e). This set of partners was modified by pathology status with increased interaction with EnC1s/EnC2s, fibroblasts and goblet/serous cells in asthma samples (Fig. 5f and Extended Data Fig. 6e).

Collectively, mast cells were abundant within proinflammatory ecosystems and were found to express remodeling mediators like *AREG*, which is involved in the regulation of tissue architecture. Inhaled corticosteroids and biologic treatment reduced the level of specific markers but did not impact aberrant tissue architecture characterized by increased cell aggregates.

## Imatinib treatment has a profound impact on lung cellular interactions

Having observed a clear mast cell signature within the inflammatory ecosystems, we next investigated how a drug designed to reduce mast cell activity impacts the cellular topography and transcriptome in the lung. We used endobronchial biopsies collected as part of a clinical trial (the KIT inhibitor in asthma (KIA) study), whereby individuals with severe refractory asthma were treated for 6 months with a tyrosine kinase inhibitor[23] (Fig. 6a, cohort 2). The primary goal of the study was to suppress mast cell activity, and the main outcomes documented were decreased airway hyperresponsiveness, as well as decreased mast cell count and reduced serum tryptase[23]. We successfully used the Xenium platform to analyze biopsies from individuals with asthma treated with placebo or imatinib (Fig. 6b and Supplementary Video 2). We detected 21 cell clusters and annotated 18 clusters based on the primary expressed genes and their localization (Fig. 6c and Extended Data Fig. 7a). Projection onto the lung cell atlas confirmed our cluster annotations (Extended Data Fig. 7b). Dominant airway cell types were present, including basal cells, ciliated cells and endothelial cells, but the expanded cluster number was mainly driven by increased immune cells: plasma cells, CD8+ T cells, monocytes, macrophages and mast cells (Fig. 6c). Samples from both placebo- and imatinib-treated participants contributed equally to these clusters within their respective treatment groups (Fig. 6c and Extended Data Fig. 7c,d).

Within the epithelial niche (Extended Data Fig. 8), alarmins and chemokines were produced by similar cell clusters as in our previous cohort (that is, basal cells, EnC2s and goblet cells; Fig. 6d). Imatinib treatment led to a dramatic reduction in the expression of all alarmins and chemokines except *CXCL8*, which was increased in goblet cells after treatment (Fig. 6d). Imatinib was particularly effective in dampening

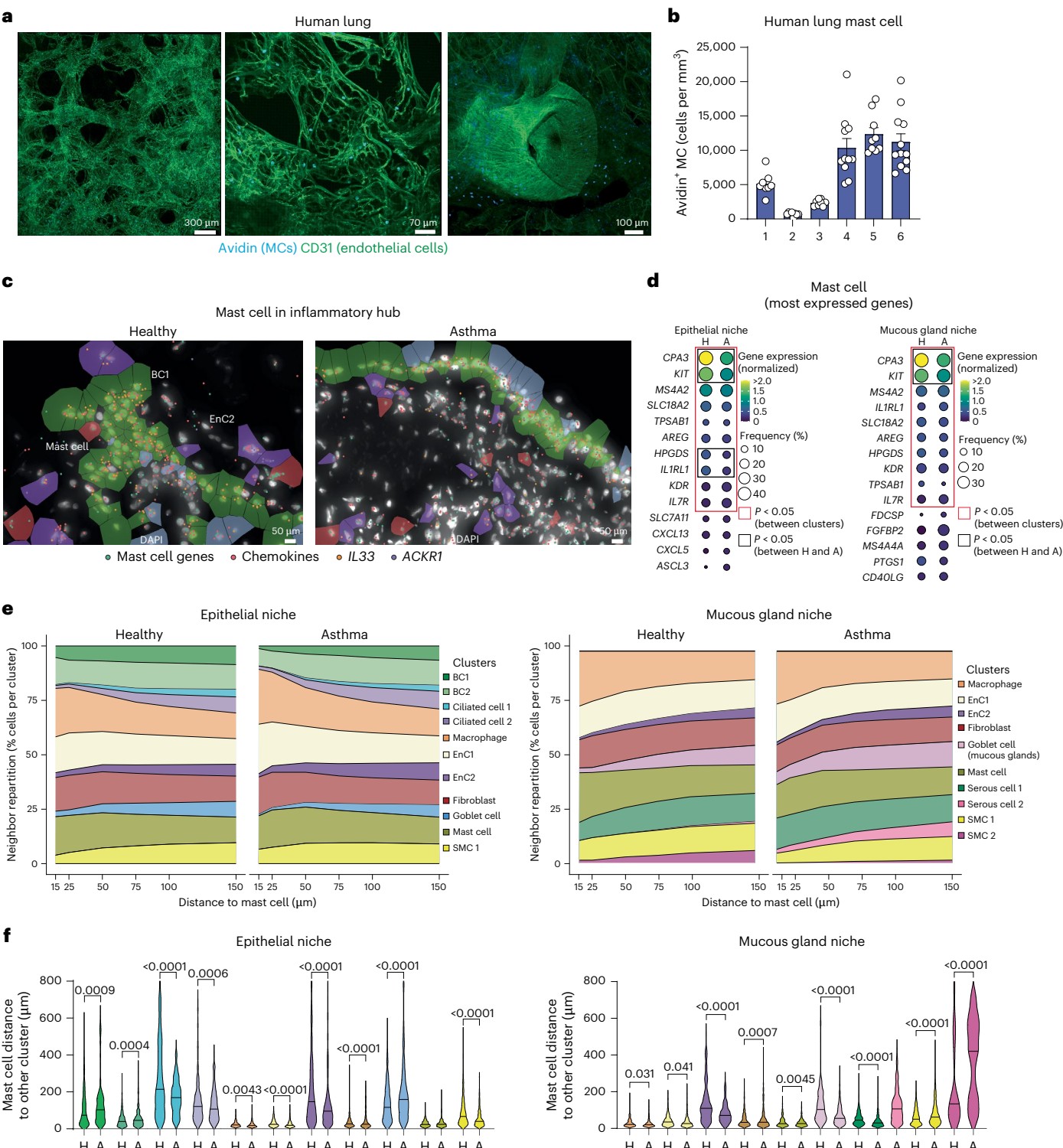

**Fig. 5 | Mast cells are regulators of proinflammatory ecosystems.**
**a**, Representative precision-cut lung slices (PCLSs) of human parenchyma showing widespread distribution of mast cells (MC). **b**, Number of mast cells normalized to the total volume of the image per donor (n = 8 (donors 1, 2 and 3), 11 (donor 4), 9 (donor 5) and 12 (donor 6) regions analyzed from n = 6 donors). Data are shown as mean ± s.e.m. **c**, Representative Xenium image of chemokine genes (red dots), *ACKR1* (purple), *IL33* (orange) and mast cell genes (cyan dots). Data are representative of two independent experiments. **d**, Gene dot plot of the

14 and 15 most expressed genes by mast cells (n = 4 donors per group, cohort 1). **e**, Repartition of neighbors along distance for mast cells (n = 4 donors per group, cohort 1). **f**, Shortest cell-to-cell distance between mast cells and other clusters (n = 4 donors per group, cohort 1). Columns alternate between healthy and asthma samples. Scales are provided for each plot. Red boxes indicate genes with a P value of <0.05 compared to other clusters, and black boxes indicate genes with a P value of <0.05 between healthy individuals and those with asthma. Data were analyzed by two-sided Mann–Whitney t-test.

the proinflammatory potential of EnC2s with an almost complete inhibition of *IL33* and *CCL19* production (Fig. 6d). We observed a particularly high level of *CCL5* produced by two clusters of CD8[+] T cells, and, although this was reduced by imatinib, residual *CCL5* expression was still observed (Extended Data Fig. 9a). Interestingly, the expression of proliferation markers was reduced across all stromal and immune clusters in individuals treated with imatinib (Extended Data Fig. 9b). Extracellular matrix genes, such as *COL3A1* and *COL1A1*, were predominantly expressed by fibroblast 1 cells, and donors treated with imatinib showed a reduced level of all types of matrix genes (Extended Data Fig. 9c). We observed that genes involved in most immune regulatory pathways, such as *LAG3* (an inhibitory receptor expressed by T cells), *CTLA4* (an inhibitory receptor expressed by T cells) and *HAVCR2* (a TIM3 inhibitory receptor expressed here by monocytes/macrophages), were all downregulated in the imatinib-treated group (Extended Data Fig. 9d).

Next, we examined the presence of inflammatory hubs defined previously (Fig. 2) and observed them in the epithelial niche associated with similar cell types (that is, basal cells and EnC2s; Fig. 6e). The inflammatory activity within these hubs was strongly reduced following imatinib treatment (Fig. 6e). We next analyzed if this was linked to any changes in transcriptional and spatial signatures. Imatinib treatment reduced the expression of most genes for basal cells and endothelial cells, but this was particularly true for *KRT15* in basal cells and *ACKR1* in EnC2s (Fig. 6f). We did not detect major changes in the cellular partners for basal cells; however, EnC2s had considerably fewer close neighbors following imatinib treatment (Fig. 6g and Extended Data Fig. 9e,f). The main clusters with reduced interaction with EnC2s were CD8[+] T cells, monocytes and plasma cells (Fig. 6g). This was associated with increased distance between cells, suggesting that cells return to their original organization (Extended Data Fig. 9e,f). In addition, imatinib treatment significantly impacted goblet cells, driving reduced expression of *CXCL6*, *MUC5B*, *UPK1B* and *KRT7* but increased expression of *CYP2F1* (Fig. 6h,i). The cellular neighbors of goblet cells were modified by imatinib treatment, and epithelial interactions were restored (that is, ciliated cells and basal cells; Fig. 6j and Extended Data Fig. 9g). Finally, although mast cells in the placebo group showed high levels of *CPA3* and *KIT* expression, individuals treated with imatinib exhibited a strong reduction in both frequency and expression of these characteristic genes, indicating the profound impact of tyrosine kinase inhibition (Fig. 6k,l). Furthermore, the frequency of mast cells near basal cells, cells within the CD8[+] T cell 2 population, fibroblasts and EnC2s was reduced in individuals treated with imatinib, leading to an overall reduction in cellular partners for mast cells (Fig. 6m and Extended Data Fig. 9h). Collectively, these data show how spatial transcriptomics can reveal mechanistic insight into the impact of drugs on cellular organization and communication during clinical trials.

## Spatial drug–target interaction reveals specialized areas of drug availability

Finally, we wanted to determine the potential of spatial transcriptomics for tailoring personalized medicine. We used the recently developed Drug2Cell tool[34] to analyze drug–target interaction using the ChEMBL database (Fig. 7a). Here, our goal was to extend the existing pipeline beyond single-cell RNA sequencing to provide spatial information for where drugs act within tissues. We first filtered drugs based on the targets (that is, genes) present in our Xenium panel (Supplementary Data 2) and explored drugs with the potential to impact our spatially resolved clusters (Fig. 2a). Interestingly, we found that recently approved tisotumab vedotin[35], which targets coagulation factor 3 (that is, F3), showed a high level of interaction with basal cells and goblet cells in individuals with asthma compared to healthy donors (Fig. 7b and Extended Data Fig. 10a). Additionally, we showed a strong signal for the monoclonal antibody to VWF caplacizumab[36], which targets endothelial cells (Fig. 7b and Extended Data Fig. 10a,b). Furthermore, we observed interesting positive signals for biologics and signaling pathway inhibitors, such as gefitinib (EGFR inhibitor), in specific cell types and regions of the lung wall (Fig. 7b). We then performed a similar analysis in the second cohort of individuals with asthma (that is, placebo + imatinib). Here, we observed similar effects of tisotumab vedotin on basal cells and caplacizumab on endothelial cells. Imatinib showed strong interaction with mast cells and fibroblasts in placebo-treated samples, and we noted a reduction in target expression in imatinib-treated samples (Fig. 7c). Restricting analysis to respiratory drugs showed that corticosteroids such as hydrocortisone were the main drugs that exhibited positive signals across multiple cell types but preferentially basal cells and endothelial cells in the epithelium and serous cells in mucous glands (Extended Data Fig. 10c). Similar observations were made in placebo- and imatinib-treated samples, with basal cells emerging as the main target for corticosteroids (Extended Data Fig. 10d). We next combined the drug information with our spatial data and represented this visually on a tissue section where these drugs are predicted to act (Fig. 7d). For example, tisotumab vedotin target signal was mainly restricted to the lung epithelium and discreet areas of the mucous glands. Conversely, hydrocortisone showed a wider effect across the bronchial wall (Fig. 7d). Similarly, imatinib, gefitinib and hydrocortisone showed specialized areas of impact in the epithelium of placebo- and imatinib-treated samples (Fig. 7e).

Collectively these data demonstrate that the combination of spatial transcriptomics with drug–target information will be a powerful tool to predict spatial therapeutic engagement at the subtissue level.

## Discussion

To meet the next challenges for pulmonary health in a future of climate change, emerging infections and an aging population, we urgently need a better understanding of the airway landscape. Here, we used

---

**Fig. 6 | Imatinib treatment disrupts cell transcriptional activity and localization in individuals with asthma. a**, Clinical trial information. **b**, Representative H&E image (left), DAPI image (middle) and cell segmentation (right). Data are representative of two independent experiments. **c**, UMAP of 36,591 cells captured with the indicated names of clusters. The image on the right shows data from donors treated with placebo (gray dots) or imatinib (red dots; cohort 2); Mono, monocytes; DC, dendritic cells; Macro, macrophages. **d**, Gene dot plot for alarmin and chemokine expression in stromal cell clusters ($n$ = 3–4 donors per group, cohort 2). **e**, Representative images showing the area analyzed and chemokine (red dots) and *IL33* (orange) transcript expression by basal cells (green cells) and endothelial cells (light brown, EnC1s; purple, EnC2s). Data are representative of two independent experiments; scale bars, 50 μm (H&E). **f**, Gene dot plot for the top 15 expressed genes in basal cells (left) and EnC1s/EnC2s (right; $n$ = 3–4 donors per group, cohort 2). **g**, Repartition of neighbors along distance for basal cells or EnC2s ($n$ = 3–4 donors per group, cohort 2). **h**, Representative image of goblet cell (green dots), *CXCL6* (red dots) and *MUC5B* (yellow dots)

transcript expression in goblet cells (blue cells). Data are representative of two independent experiments. **i**, Gene dot plot of the 15 most expressed genes by goblet cells ($n$ = 3–4 donors per group, cohort 2). **j**, Repartition of neighbors along distance for goblet cells ($n$ = 3–4 donors per group, cohort 2). **k**, Representative image of mast cells (dark brown) and endothelial cells (light brown, EnC1s; purple, EnC2s) and transcripts for mast cells (brown dots), *CPA3* (red dots) and EnC (purple dots) genes. Data are representative of two independent experiments. **l**, Gene dot plot of the ten most expressed genes by mast cells ($n$ = 3–4 donors per group, cohort 2). **m**, Repartition of neighbors along distance for mast cells ($n$ = 3–4 donors per group, cohort 2). Columns alternate between samples from donors treated with placebo (P) or imatinib (I). Scales are provided for each plot. Red boxes indicate genes with a $P$ value of <0.05 compared to other clusters, and black boxes indicate genes with a $P$ value of <0.05 between placebo and imatinib. Data were analyzed by two-sided Mann–Whitney $t$-test.

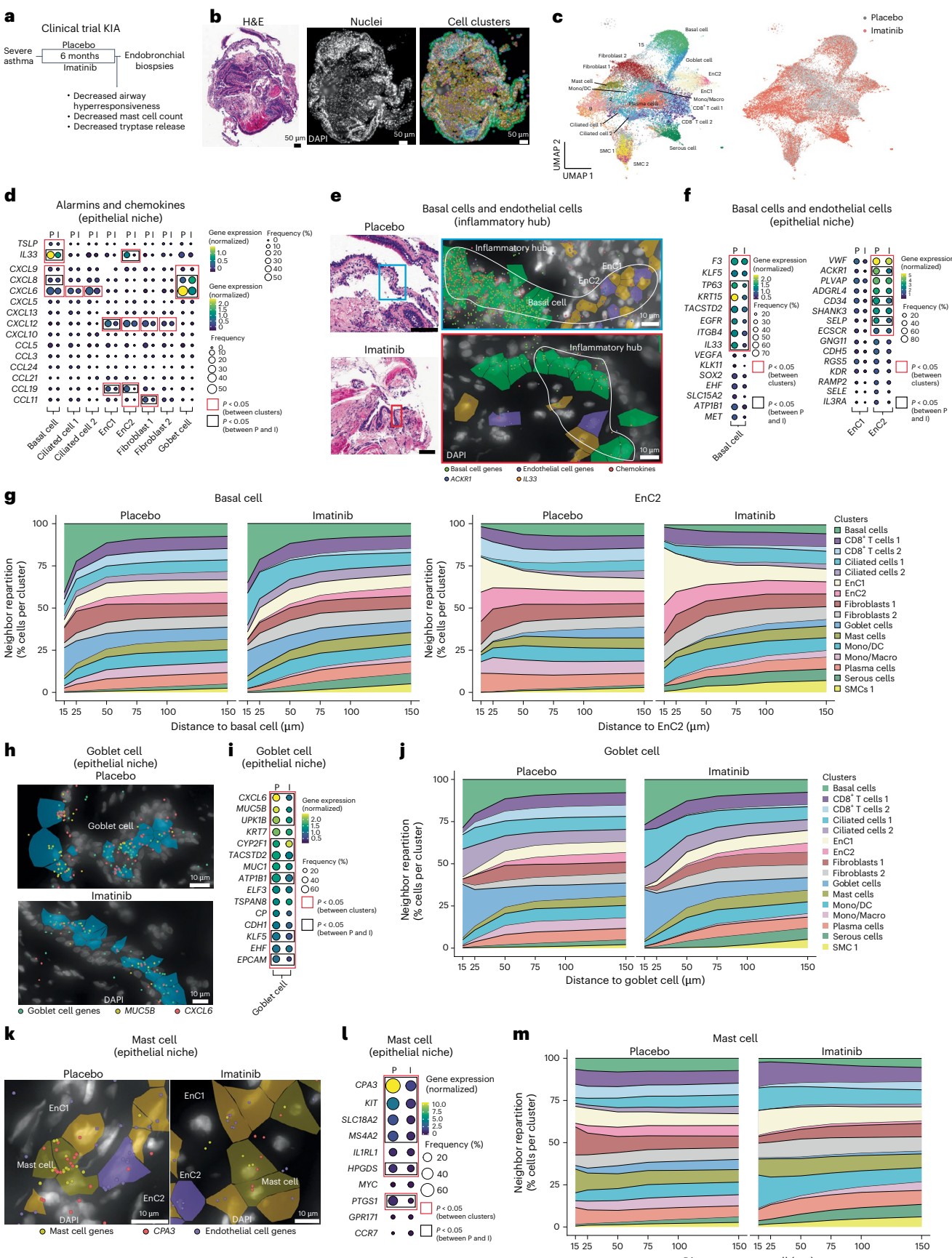

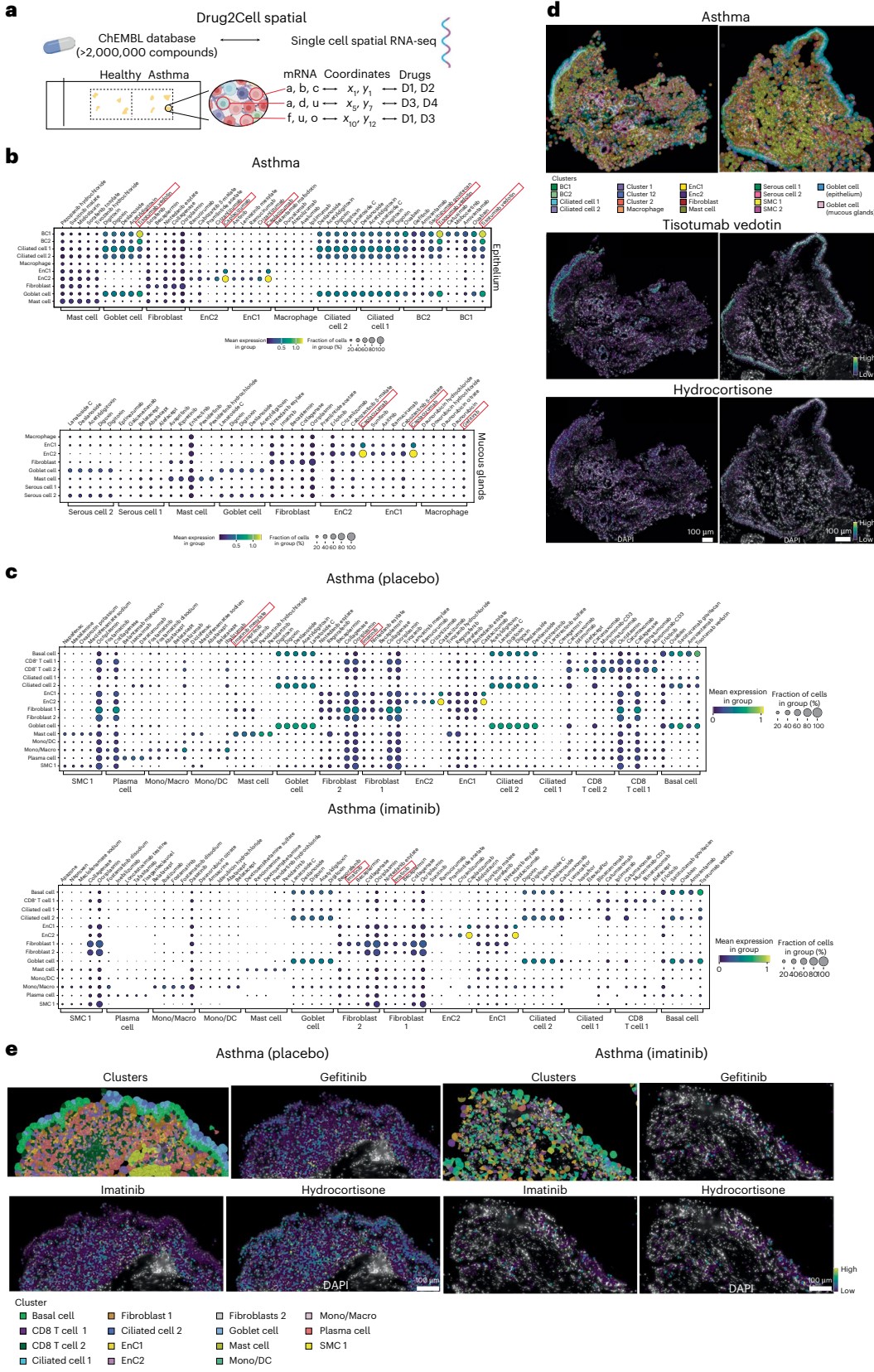

**Fig. 7 | Spatial Drug2Cell analysis in lung biopsies. a**, Schematic depicting the principle of the analysis combining Xenium data and data from the ChEMBL database. **b**, The top five drugs per cluster with the strongest interaction and frequency of cells impacted in the epithelium and mucous gland areas of individuals with asthma (cohort 1). **c**, The top five drugs per cluster of placebo- and imatinib-treated epithelial ROIs (cohort 2). **d**, Spatial representation of cellular targets for tisotumab vedotin and hydrocortisone in two asthma biopsies. The color scale indicates strength of interaction. **e**, Spatial visualization of cellular targets for imatinib-, gefitinib- and hydrocortisone-treated biopsies. The color scale indicates interaction strength. For drug dot plots, the size of the dot reflects the frequency of cells within the cluster impacted by the drug, and color indicates the interaction strength. Illustrations created with Adobe Illustrator and Biorender.com.

single-cell spatial transcriptomics to investigate the lung bronchial wall, the primary site of tissue inflammation and remodeling in many chronic lung disorders[5,37]. New approaches relying on nondissociative tissue methods combined with quantitative imaging represent powerful tools to answer challenging questions related to human biology. Indeed, our data are in agreement with reports from the lung cell atlas[8,38–40], confirming specialized populations of cells in specific areas of the lungs. Spatial niches, such as bronchi and associated adventitial areas, have recently gained increasing attention, garnering the realization that location influences cell identity and functional phenotype[41–44]. Here, we successfully identified ~20 cell clusters; however, important cell types such as eosinophils are missing[25]. This is mainly due to the low mRNA transcript abundance of these cell types and the cell segmentation approaches currently available[45,46].

We have described specialized lung neighborhoods, enriched in chemokines and alarmins expressed by specific cell types. These 'cellular communities' present in healthy donors as well individuals with asthma lead to the formation of ecosystems that serve as potential hot spots for immune cell recruitment and regulation[42,44]. This idea of discrete cellular ecosystems is reinforced by the presence of molecular interactions defined by ligand–receptor pairings, such as IL-33+ basal cells and EnC2s and ST2+ mast cells and AREG+ mast cells interacting with EGFR+ BC1s or chemokines and ACKR1+ expressed by EnC2s. Interestingly, we found mosaic expression of *ACKR1* in endothelial cells, with EnC2s expressing the highest level linked with chemokines and IL-33 secretion. ACKR1 is known to transport and present chemokines essential for leukocyte trafficking[47–49]. Our data indicate that ACKR1 regulates the organization of these cellular communities by modulating the availability of chemokines and therefore influencing the retention or clearance of leukocytes. Interestingly, ACKR1 is a known factor associated with increased IgE and asthma[50], but its specific role in asthma is unclear[51,52].

Mast cells were among the most abundant of all immune cells identified and expressed genes facilitating the regulation of immune ecosystems (for example, *AREG* and proteases). Our data clearly indicate that mast cells are central to the monitoring of inflammatory niches, and expression of *AREG* was maintained in asthma biopsies despite treatment[33,53]. Mast cell proteases have been shown to impact lung bronchial epithelium by disrupting their morphology[54,55], and here, the strong inhibition of *CPA3* expression by imatinib may help restore barrier integrity during asthma.

Surprisingly, we observed a reduction in the expression of inflammatory chemokines in all our samples from donors with asthma[1,2]. This is in accordance with other datasets where, for example, the expression of *IL33* and *CXCL6* was decreased[11]. As shown for IL-33, the expression and maturation of this key alarmin is still an active area of research[56]. Moreover, despite the reduction in mRNA levels, the spatial association of cells during asthma remodeling may lead to an intense 'hotspot' of production, leading to higher levels of chemokines and alarmins locally. Furthermore, these inflammatory ecosystems were found to contain a variety of components dysregulated during asthma, such as mucous production (that is, *MUC5B*). Increased MUC5AC coupled with a decrease in MUC5B expression has been described during asthma[57,58]. The very high expression of mucous genes suggests that current treatments do not tackle this crucial facet of asthma pathology[59]. A vital aspect of our study is the potential for using historical clinical trial samples to investigate aspects of therapeutic intervention, beyond the anticipated primary target or clinical outcome. Furthermore, our improved pipeline to analyze drug–target interactions indicates how to garner information from omics technologies for therapeutic development, drug repurposing and personalized medicine.

Collectively, our data reveal an airway wall marked by discrete regulatory cellular ecosystems formed by chemokines and alarmins secreted by unique combinations of stromal cells. During chronic inflammation, the transcriptional profiles and spatial organization of these inflammatory hubs are disrupted, leading to altered intercellular communication and thus development of pathology.

## Online content

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

## Methods

### Antibodies and probe
Alexa Fluor 488-conjugated anti-human CD31 (clone WM59) was from Biolegend, and avidin (A887) conjugated to Alexa Fluor 647 (A20186) was from Thermo Fisher.

### Human donors
A summary of the clinical data is available in Supplementary Table 1. Briefly, adults (18–65 years old) with severe asthma, as defined by European Respiratory Society and American Thoracic Society guidelines, were recruited. All had a body mass index of 18.5–35 kg m$^{-2}$, were nonsmokers or had not smoked for at least the past 12 months (with a less than 10 pack year history) and had a forced expiratory volume in 1 s of ≥1 l ≥60% predicted. For participant stratification between mild and severe asthma, severe asthma (n = 4) was defined by the European Respiratory Society and American Thoracic Society consensus criteria, and all participants mapped to Global Initiative for Asthma (GINA) 5 (track 2) of the current GINA 2024 guidelines. Participants with mild asthma were GINA steps 2–3 on either low-dose inhaled corticosteroids (ICS) (1/3) or low-dose ICS/long acting beta2 agonist (LABA) (2/3). The collection of endobronchial biopsies and clinical information was approved by London–Bloomsbury Research Ethics Committee under approval number REC 19/LO/1675.

Donor information for participants treated with placebo or imatinib, including demographic and lung function results, can be found in the original publication[23] and clinical trial data (ClinicalTrials. gov NCT01097694). A summary of donor information is provided in Supplementary Table 1 and Supplementary Data 1.

Human adult samples used in this research project were obtained from the Imperial College Healthcare Tissue Bank (ICHTB). ICHTB is supported by the National Institute for Health Research (NIHR) Biomedical Research Centre based at Imperial College Healthcare NHS Trust and Imperial College London. ICHTB is approved by Wales REC3 to release human material for research (17/WA/0161), and the samples for this project (R22006) were issued from subcollection reference number ICB_NC_21_017. The biopsy samples from University Hospitals of Leicester were issued under MTA 2021S-0809-2029 between University of Leicester/UHL and Imperial College London and were approved by the research ethics committee (MREC: 08/H0406; IRAS: 8824).

### Tissue collection
Endobronchial bronchoscopies were performed at the Royal Brompton Hospital. Biopsies were preserved in cell medium (DMEM; Thermo Fisher), quickly transferred into 4% paraformaldehyde for 1 h at room temperature or 10% neutral buffered formalin for 24 h at room temperature (GeoMX samples) and finally maintained in 70% ethanol until embedding in paraffin.

### Spatial single-cell transcriptomics
Tissue sections (5 μm) from formalin-fixed paraffin-embedded (FFPE) blocks were placed on a Xenium slide. Additional sections were placed on normal microscopic slides (Thermo Fisher) for H&E staining. Up to eight biopsies were placed on a single Xenium slide after careful tissue alignment. For the KIA samples, tissue sections were deparaffinized using xylene for ~10 min at room temperature. A drop of DPX mounting reagent (Merck) was placed on top of each section and left to dry at room temperature overnight. Using a razor blade, sections were gently lifted off the slide and transferred to the Xenium slide. Up to nine transferred sections were placed on the Xenium slide, and the slide was left in a slide oven for 4 h at 56 °C. An additional control FFPE section was placed with the 'transferred' sections as a control for mRNA integrity. Next, Xenium slides were processed by 10x Genomics for the rest of the experiment according to the manufacturer's protocols. The 339 genes analyzed are shown in Supplementary Data 2 and include the predesigned 'lung' panel from 10x Genomics plus 50 manually selected additional genes.

### Single-cell spatial transcriptomics analysis
Raw data were processed by the 10x Catalyst team using their publicly available pipeline. In summary, we detected 106,304 cells across the two slides with 3,566,174 transcripts and a median of 31 and 15 transcripts per cell for the samples collected at Imperial College and KIA samples, respectively. Low-quality transcripts (that is, a q value of <20) and cells with no transcripts were removed from the analysis pipeline. Transcript counts were normalized using a negative binomial regression (sctransform v0.4.1) in the Seurat (v5.2.1) pipeline[60], and the frequency of cells expressing a specific transcript was calculated using the raw transcript count. We used the cell segmentation provided by the onboard analysis software (Xenium Explorer version 2.0) based on nuclear expansion. We first performed an unbiased clustering using the onboard Xenium Explorer parameter, which relies on graph-based clustering[61]. This algorithm consists of building a sparse nearest neighbor graph followed by Louvain modularity optimization (https://www.10xgenomics.com/support/software/cell-ranger/latest/algorithms-overview/cr-gex-algorithm). We identified 18 and 21 clusters in samples collected at Imperial College and KIA samples, respectively. Clusters were manually annotated using the highly expressed genes, projected on publicly available lung datasets (https://5locationslung.cellgeni.sanger.ac.uk/all)[8] and visualized in situ using Xenium Explorer (version 3.1.1, 10x) to ascertain their accuracy. Example images, including cell segmentation and transcript heat maps, were extracted from Xenium Explorer. To separate epithelial and mucous gland areas, we manually selected the regions on Xenium Explorer and carefully avoided the area with disrupted tissue organization. These areas were compared with H&E slides for further confirmation of the accuracy of selection. Transcripts and cell coordinates were extracted and analyzed as indicated in relevant sections. The following R packages were used in RStudio (RStudio 2023.06.1) for analysis and representation of the data: umap (v0.2.10.0) and ggplot2 (v3.5.1).

### Cellular neighbor analysis
To perform nearest neighbor analysis, we used the phenoptr tool[62]. Cellular coordinates (that is x and y cell position based on center of the segmented cell) were extracted from Xenium outputs and analyzed for their shortest distance to other clusters. Distances to other clusters were represented as a histogram showing frequency and distance until 100 μm or frequency of specific clusters were analyzed for their close neighbors (that is, less than 50 μm). As a threshold, we used the following parameter: at least one cell from each cluster being compared needed to be within the defined radius to count as a positive interaction (that is, at least one mast cell and one fibroblast needed to be within 50 μm of each other to be recorded as a positive interaction).

### Drug-to-cell spatial transcriptomics
Drug2Cell integrates user-provided single-cell expression data with drug–target interactions from the ChEMBL database (https://www.ebi.ac.uk/chembl/) to comprehensively evaluate drug target expression at single-cell resolution. The Drug2Cell Python analysis pipeline was adapted from the publicly available Drug2Cell Python pipeline (Drug2Cell version 0.1.0) published by the Teichmann lab[34] for spatial transcriptomics. Briefly, drugs and targets were obtained from the ChEMBL database (version 30) and filtered based on the genes analyzed (Supplementary Data 3). We then calculated a score for each drug based on the level of raw gene expression and followed the statistical approach described in the original publication. For spatial representation, we combined the score for each cell with the coordinates calculated using the Xenium platform, extracted a spatial matrix color, coded with the strength of interaction using SquidPy (version 1.4.1) and manually superimposed this onto images of a DAPI-stained Xenium section with the spatial matrix for specific drugs. Details on the Drug2Cell Python package can be found at GitHub (https://github.com/Teichlab/drug2cell).

## GeoMx spatial transcriptomics

Detailed experimental methods (NanoString) are described in Zimmerman et al.[63]. FFPE human lung samples were baked overnight at 37 °C, followed by 3 h of baking at 65 °C, and loaded onto a Leica Bond RX Fully Automated Research Stainer for subsequent processing steps. The processing protocol included three major steps: (1) slide baking, (2) antigen retrieval for 20 min at 100 °C and (3) treatment with Proteinase K (1.0 μg ml⁻¹ in PBS) for 15 min. Following these steps, slides were removed from the Leica Bond RX, and a cocktail of GeoMx Full Transcriptome Atlas probes was applied to each slide and allowed to hybridize at 37 °C overnight in a humid chamber. The following day, slides were washed, blocked and allowed to incubate with a combination of Alexa Fluor 488-labeled anti-α-SMA (Invitrogen/Thermo, 53-9760-82, clone 1A4), Alexa Fluor 594-labeled anti-vimentin (Santa Cruz, sc-373717 AF594, clone E-5), Alexa Fluor 647-labeled anti-CD45 (Cell Signaling Technology, 13917BF, clone D9M8I) and Syto83 nucleic acid stain. Slides were stained for 1 h at room temperature in a humid chamber, washed and loaded onto a GeoMx instrument. On the GeoMx machine, slides were scanned for fluorescence, and ROIs were collected from the following areas: smooth muscle, epithelium and subepithelium. The GeoMx device exposed ROIs to 385-nm light (UV), releasing the indexing oligonucleotides. Indexing oligonucleotides were collected with a microcapillary and deposited into a 96-well plate. Samples were dried down overnight and resuspended in 10 μl of DEPC-treated water. PCR was performed using 4 μl of each sample, and the oligonucleotides from each ROI were indexed using unique i5 and i7 dual-indexing systems (Illumina). PCR reactions were purified twice using AMPure XP beads (Beckman Coulter) according to the manufacturer's protocol. Purified libraries were sequenced on an Illumina NovaSeq 6000. Data analysis was performed as previously described[63]. Following removal of targets consistently below the limit of quantification (that is, <5,000 raw reads) and negative probes, the limit of detection above which a gene was called 'detected' was defined as 2 s.d. above the geometric mean of negative probes. Datasets were normalized using upper quartile (Q3) normalization. Data analysis was then performed using the DSP platform and R software. Cell deconvolution analysis was performed using the following R packages: GSVA (v2.0.1), Stringr (v1.5.1) and Dplyr (v1.1.4)[20].

## PCLSs

The PCLS model was adapted from previously described protocols[31,64,65]. Human lung tissue and pulmonary vessels were collected from anonymized donors at Hammersmith and Royal Brompton hospitals NHS trusts and immediately fixed in 4% paraformaldehyde overnight at 4 °C. Following fixation, 100- to 200-μm sections were prepared using a Compresstome VF-300 vibrating microtome (Precisionary Instruments).

PCLSs were permeabilized in PBS supplemented with 0.5% Triton (Sigma) for 1 h at room temperature and blocked in animal-free blocker (2BSCIENTIFIC) for 1 h. Slices were incubated with the indicated primary antibodies and avidin overnight at 4° C in 25% animal-free blocker in PBS, and, where required, PCLSs were incubated with secondary antibodies for 4 h at room temperature in 25% animal-free blocker in PBS. Lung slices were mounted on microscope slides (Thermo Fisher), immersed in ProLong Diamond (Thermo Fisher) and kept at 4 °C until image acquisition.

## Image acquisition

Images were acquired on a LEICA SP4 or SP8 (LEICA) using a ×20/0.7-NA (SP4), ×20/0.75-NA (SP8) or ×10/0.4-NA (SP4 and SP8) objective with a resolution of 512 × 512 or 1,024 × 1,024 pixels. Motorized stages were used for tile scan imaging and merged using LEICA built-in software (LAS, version 5.1.0) with a 10% overlap threshold. H&E images of biopsies were acquired on a Aperio VERSA 8 with a ×20 objective and analyzed using ImageScope (Leica, v12.4.6.5003)

## Image analysis

Image analysis and rendering were performed using IMARIS 10.1 (Andor, Bitplane). LIF files were converted into IMARIS .ims files using IMARIS Converter software v10.1 (Andor, Bitplane). The number of mast cells was determined using the semiautomatic spot function (cell diameter of ~10 μm for mast cells). Numbers of cells were normalized to the total volume of the image. Thresholds for cell analysis in PCLS sections were maintained across the experimental conditions to provide accurate comparisons. All image analysis was performed on raw images without fluorescence modification.

## Statistical analysis

Most statistical analyses were performed using Prism software (GraphPad v9.4.1), R (v4.3.0), RStudio (v2023.06.1) or Python (v3.12). The results are expressed as mean or mean ± s.e.m., and the $n$ numbers for each dataset are provided in the figure legends. Comparisons between two groups were performed using a paired or unpaired Student's or Mann–Whitney $t$-test as appropriate. Statistical significance was accepted at $P < 0.05$.

## Reporting summary

Further information on research design is available in the Nature Portfolio Reporting Summary linked to this article.

## Data availability

The analysis was performed using published and freely available software and code mentioned in relevant sections in the Methods. Raw and processed data are available at the NIH Gene Expression Omnibus online database under accession number GSE269354. The ChEMBL database used for Drug2Cell spatial analysis is available at https://ftp.ebi.ac.uk/pub/databases/chembl/ChEMBLdb/latest/. Single-cell RNA-sequencing data are available on the Sanger Institute database (https://5locationslung.cellgeni.sanger.ac.uk/all). Nanostring raw data and Q3 norm data are available in Supplementary Data 4 and 5. Further information and requests for resources and reagents should be directed to and will be fulfilled by lead author R.J. (r.joulia@imperial.ac.uk). Source data are provided with this paper.

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

## Acknowledgements

This work was supported by funds from the Wellcome Trust (107059/Z/15/Z and 220254/Z/20/Z) to C.M.L. R.J. is supported by fellowships from the British Heart Foundation Imperial College Centre for Research Excellence (RE/18/4/34215), ACTERIA EFIS Allergology 2023 and Wellcome Trust (227236/Z/23/Z). The Facility for Imaging by Light Microscopy at Imperial College is supported,

in part, by funding from the Wellcome Trust (104931/Z/14/Z). This project was supported by the NIHR Imperial Biomedical Research Centre. The views expressed are those of the authors and not necessarily those of the NIHR or the Department of Health and Social Care. We would like to thank the Catalyst team from 10x Genomics for their invaluable help and for allowing us to try our risky protocols, C. Preston for histology expertise, H. Katz for the KIA samples, the ICAN Asthma Network and M. Haniffa for support on data analysis and the Drug2Cell pipeline.

## Author contributions

Conceptualization: R.J. and C.M.L. Methodology: R.J. Investigation: R.J. and S.P. Resources: L.L., W.J.T., L.Y., S.A.W., F.P., M.A.-S., S.S., K.N.C., J.L., J.A.B. and E.I. Formal analysis: R.J. and S.P. Writing, original draft: R.J. and C.M.L. Writing, review and editing: R.J., S.P., W.J.T., L.L., L.Y., K.N.C., S.S. and C.M.L. Supervision: R.J. and C.M.L.

## Competing interests

The authors declare no competing interests.

## Additional information

**Extended data** is available for this paper at https://doi.org/10.1038/s41590-025-02161-3.

**Correspondence and requests for materials** should be addressed to Régis Joulia or Clare M. Lloyd.

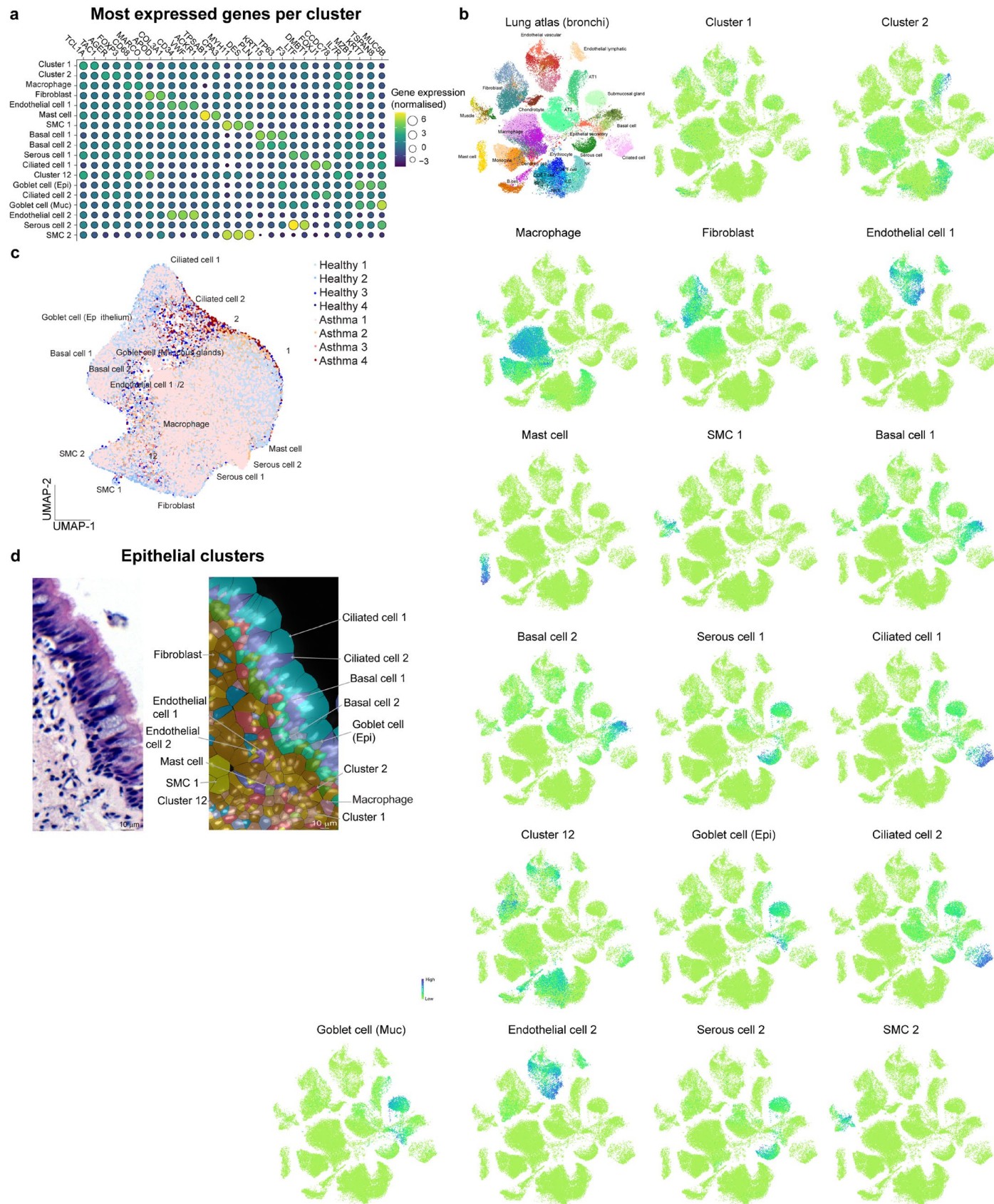

**Extended Data Fig. 1 | Identification of clusters generated by Xenium and projection on lung cell atlas. a**, Gene dot plot showing the top 2 genes expressed per cluster across all donors, size and color indicate level of expression (cohort #1). **b**, UMAP projection of lung cell atlas and expression level of the top 5 genes expressed by each cluster. Data available at (https://5locationslung.cellgeni. sanger.ac.uk/all). **c**, UMAP projection showing each donor (healthy and asthma). **d**, Corresponding H&E staining and Xenium identified clusters at the epithelium showing the cellular diversity identified and spatial organisation. Scale bars, 10 μm (representative of 2 independent experiments).

## Region of interest separation

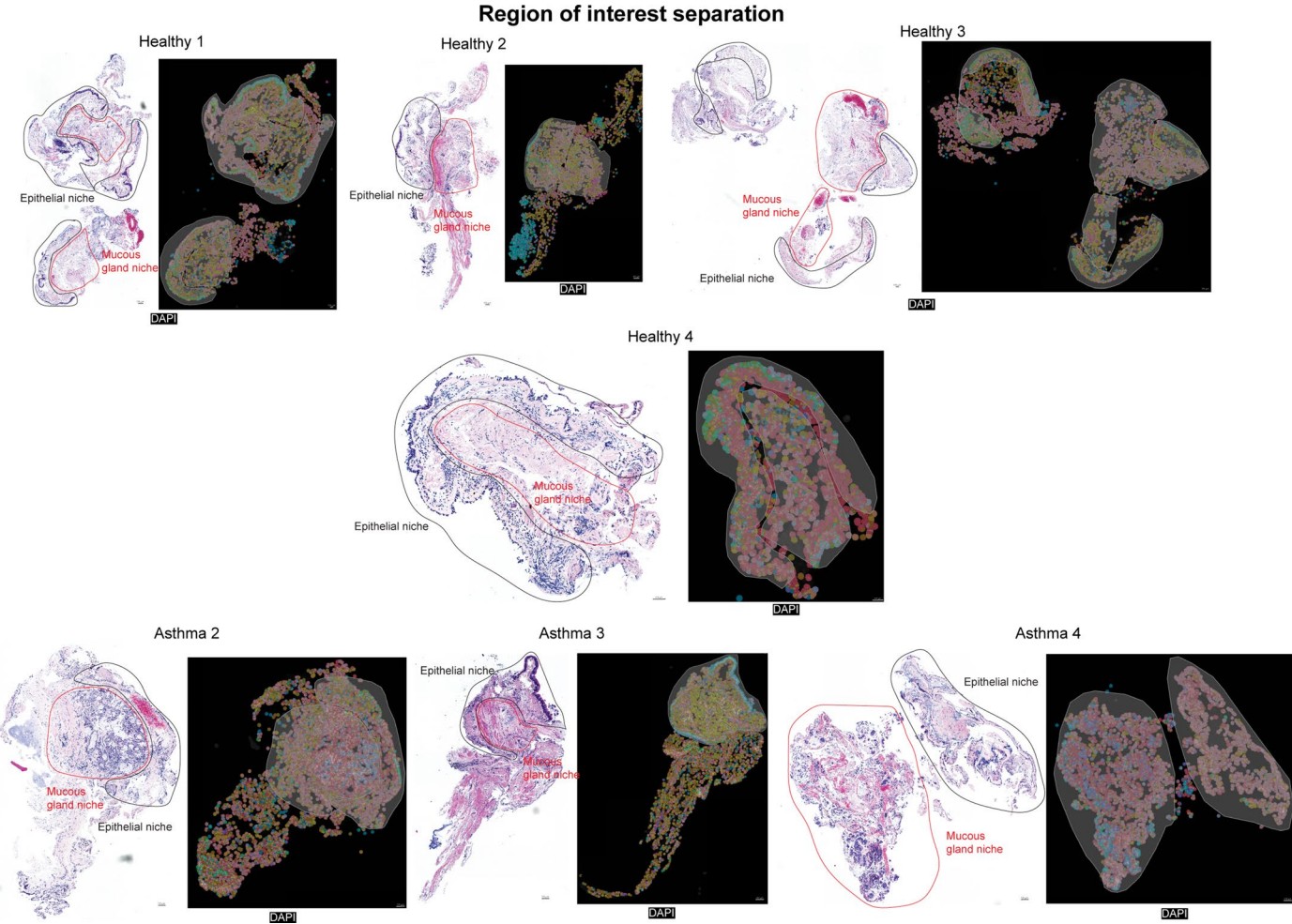

**Extended Data Fig. 2 | Region of interest selection across all healthy and asthma biopsies.** For each biopsy, H&E and Xenium images are shown, and regions used for epithelial and mucous gland downstream analysis are represented (cohort #1). Scale bars, 100 μm (representative of 1 independent experiment).

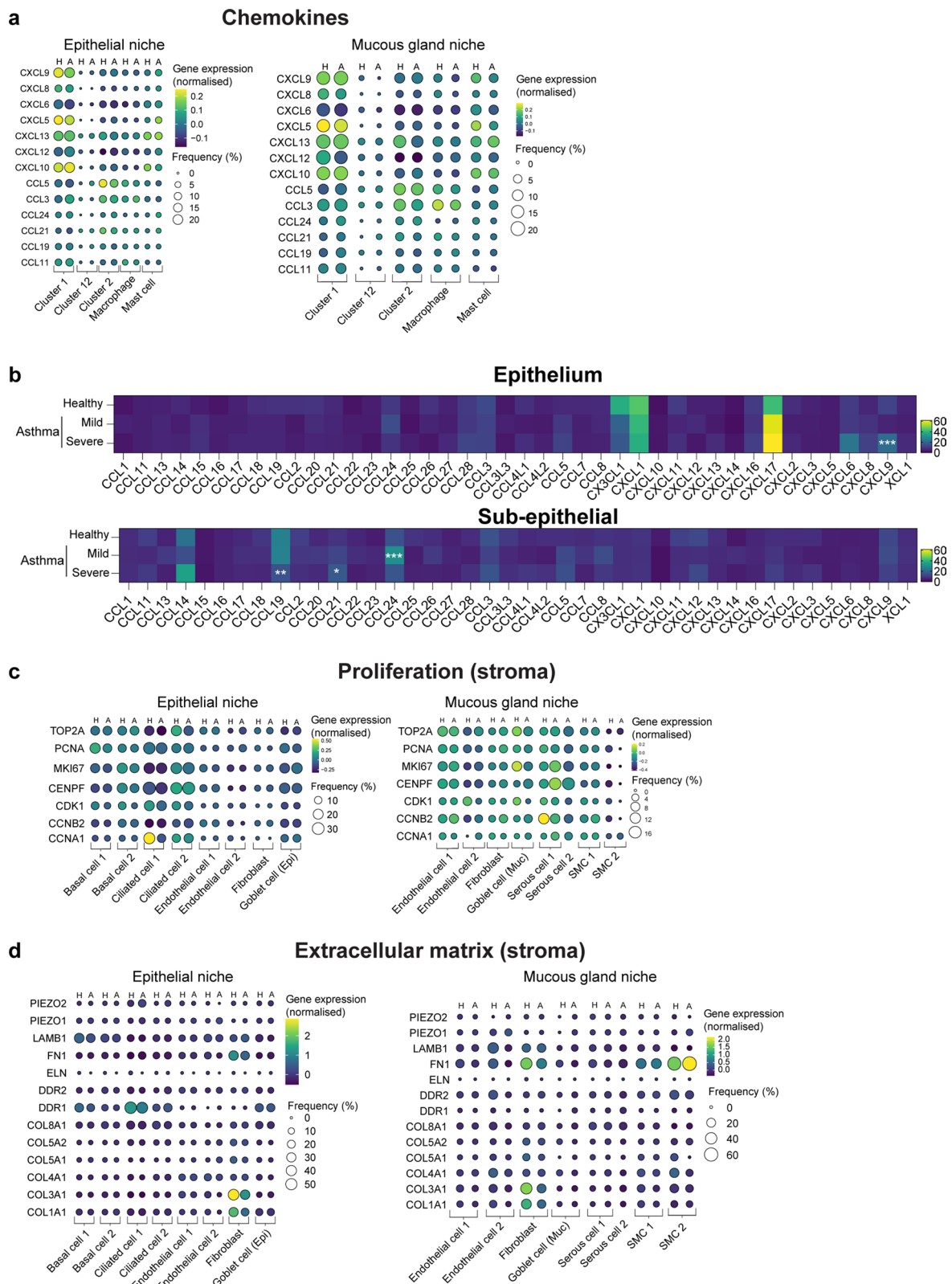

**Extended Data Fig. 3 | Epithelial and mucous gland niches chemokine, proliferation, extracellular matrix, goblet cell. a,** Gene dot plot for chemokine expression in immune and undefined cell clusters (cohort #1). **b,** Heat map projection of chemokines expression in epithelial and sub-epithelial ROIs (4-9 ROIs per group from n = 4 donors per group, cohort #3). **c,** Gene dot plot for proliferation markers expression in stroma cell clusters (cohort #1). **d,** Gene dot plot for extracellular matrix gene expression in stroma cell clusters (cohort #1).

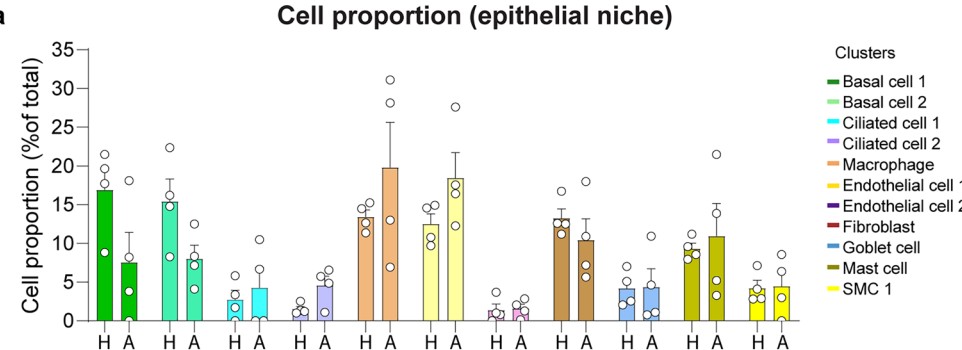

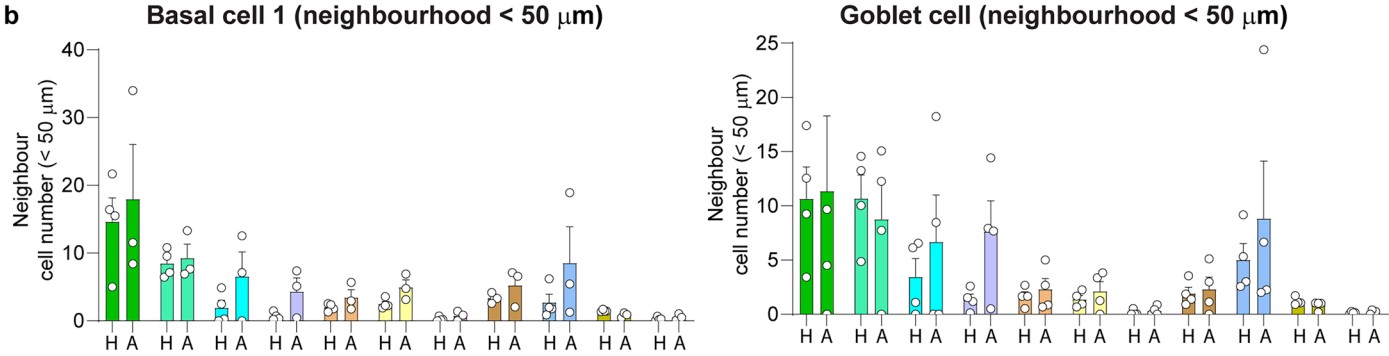

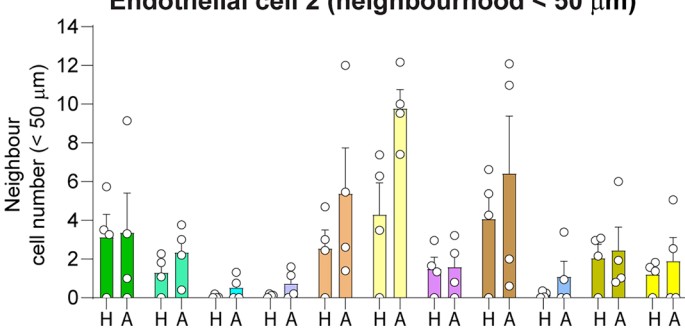

**Extended Data Fig. 4 | Spatial parameters of epithelial niche. a**, Cell proportion for each cluster of the epithelial niche in healthy (H) and asthma (A) samples (n = 4 donors per group, cohort #1). **b**, Average neighboring cell number between basal cell 1, goblet cell and endothelial cell 2 and each cluster (n = 4 donors per group, cohort #1). Mean ± SEM.

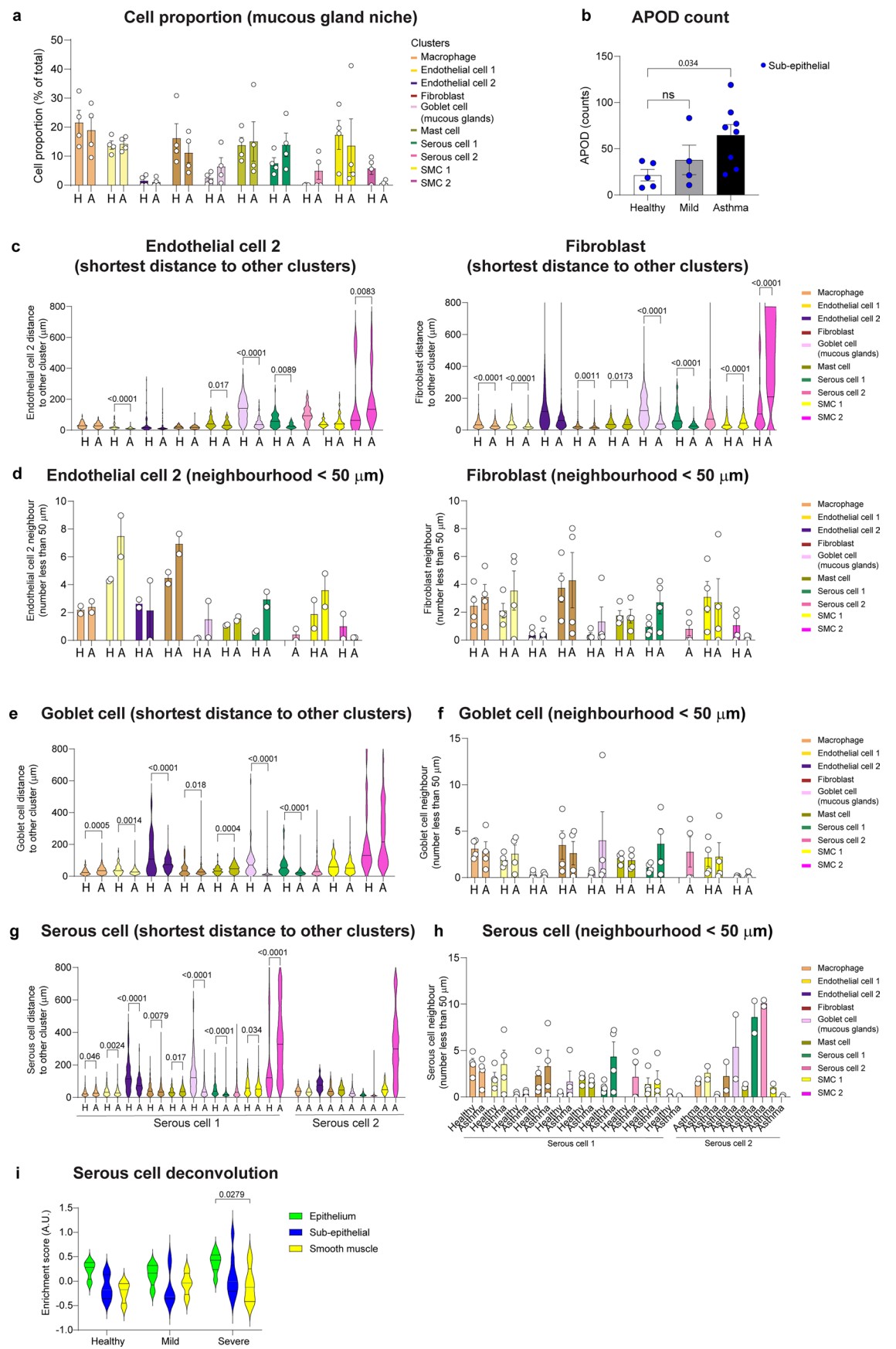

**Extended Data Fig. 5 | See next page for caption.**

**Extended Data Fig. 5 | Mucous glands area spatial profile. a**, Cell proportion for each cluster of the epithelial niche in healthy (H) and asthma (A) samples (n = 4 donors per group, cohort #1). Mean ± SEM. **b**, APOD count in sub-epithelial ROIs (n = 5 (healthy), 4 (mild) and 8 (severe) ROIs per group from n = 4 donors per group, cohort #3). Mean ± SEM. **c**, Shortest distance between endothelial cell 2 and fibroblast to each cluster (pooled cells n = 4 donors per group, cohort #1). **d**, Average neighbouring cell number between endothelial cell 2 and fibroblast to each cluster (n = 4 donors per group, cohort #1). Mean ± SEM **e**, Shortest distance between goblet cell to each cluster (pooled cells n = 4 donors per group,

cohort #1). **f**, Average neighbouring cell number between goblet cell to each cluster (n = 4 donors per group, cohort #1). Mean ± SEM. **g**, Shortest distance between serous cells to each cluster (pooled cells n = 4 donors per group, cohort #1). **h**, Average neighbouring cell number between serous cell to each cluster (n = 4 donors per group, cohort #1). Mean ± SEM. **i**, Serous cell enrichment score (epithelium ROIs n = 4 (healthy), 5 (mild) and 7 (severe); sub-epithelial ROIs n = 6 (healthy), 4 (mild) and 5 (severe); smooth muscle ROIs n = 4 (healthy), 4 (mild) and 9 (severe) from n = 4 donors per group, cohort #3). Two-sided Mann-Whitney t test.

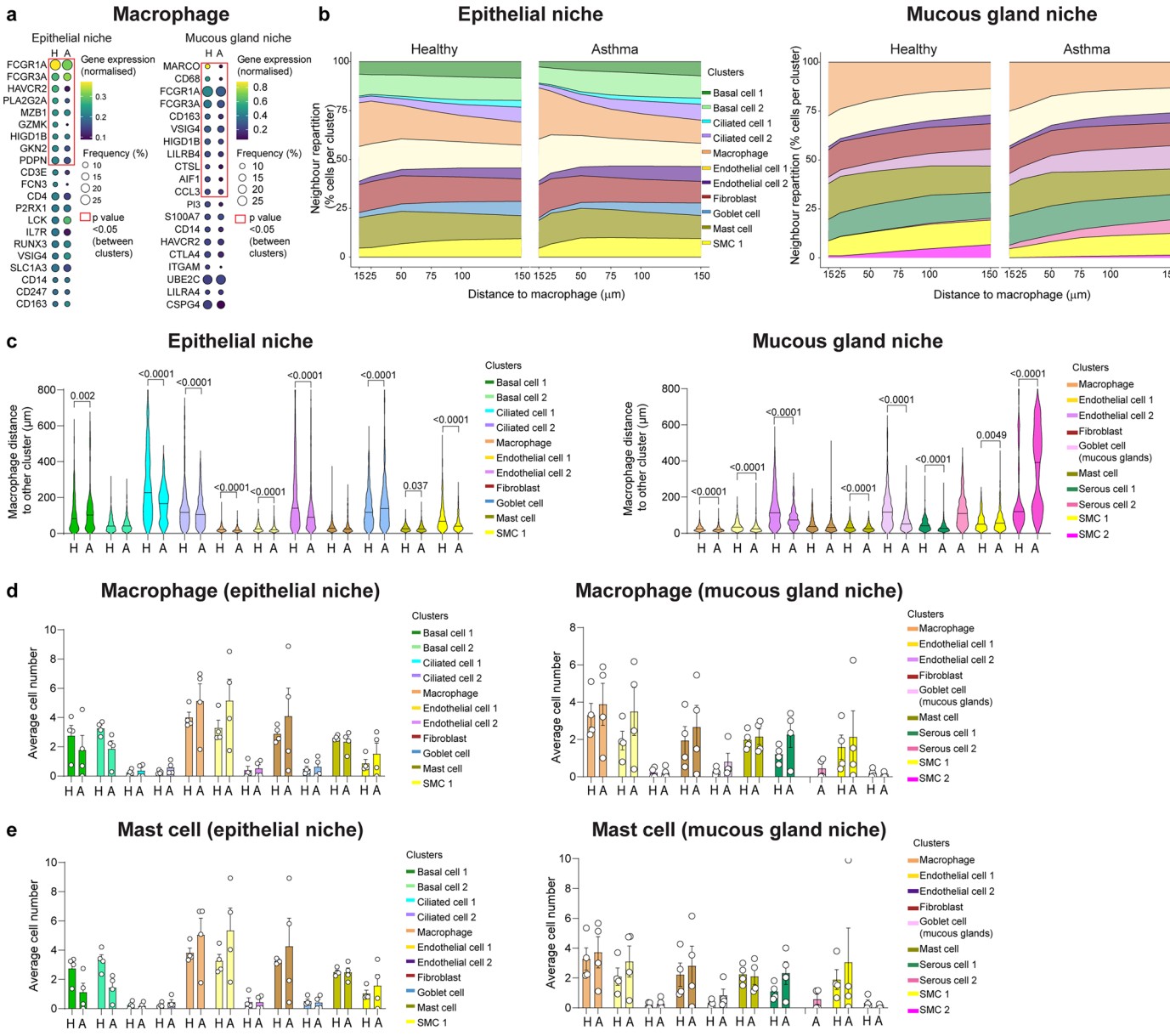

**Extended Data Fig. 6 | Macrophage spatial profile and mast cell neighbourhood. a**, Gene dot plot for top 20 expressed genes in macrophage cells in the epithelial and mucous gland niche (pooled cells from n = 4 donors per group, cohort #1). **b**, Repartition of macrophage neighbours in the epithelial and mucous gland niche (pooled cells from n = 4 donors per group, cohort #1). **c**, Shortest distance between macrophage to each cluster in the epithelial and mucous gland niche (pooled cells n = 4 donors per group, cohort #1). **d**, Average neighbouring cell number between macrophage to each cluster in the epithelial

and mucous gland niche (n = 4 donors per group, cohort #1). Mean ± SEM. **e**, Average neighbouring cell around mast cells in the epithelial and mucous gland niche (n = 4 donors per group, cohort #1). Mean ± SEM. Columns alternate between healthy (H), and asthma (A) samples. Scales are provided for each plot, red boxes indicate genes with p value < 0.05 compared to other clusters, black boxes indicate genes with p value < 0.05 between healthy and asthma. Two-sided Mann-Whitney t test.

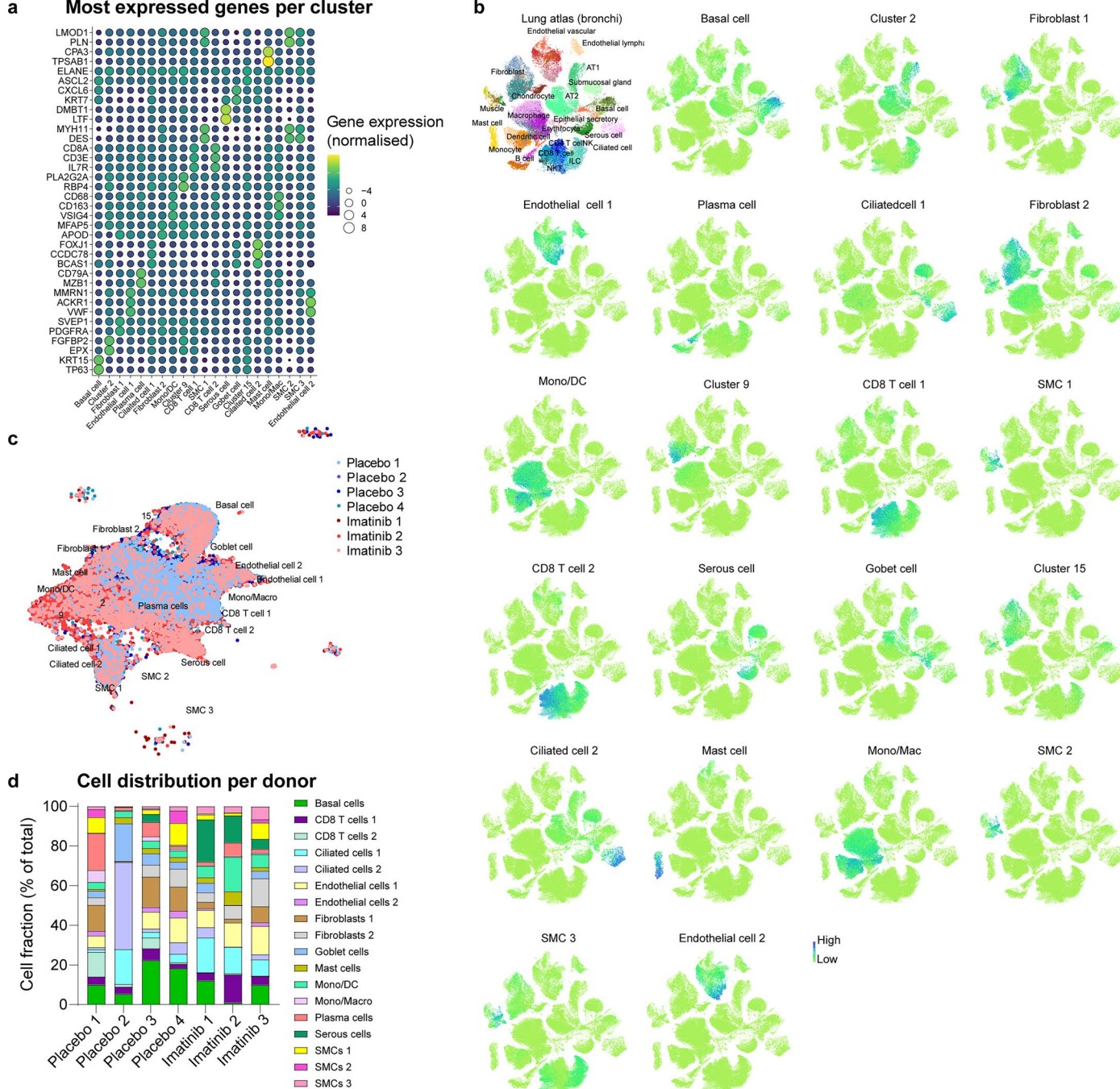

**Extended Data Fig. 7 | Identification of clusters in placebo and imatinib treated samples and projection on lung cell atlas. a**, Gene dot plot showing the top 2 genes expressed per cluster across all placebo and imatinib samples, size and colour indicate level of expression (cohort #2). **b**, UMAP projection of lung cell atlas and expression level of the top 5 genes expressed by each cluster. **c**, UMAP projection showing each donor (placebo and imatinib). **d**, Cell fraction of each cluster per donor.

## Region of interest separation

**Extended Data Fig. 8 | Region of interest selection in placebo and imatinib treated samples.** For each biopsy, H&E and Xenium images are shown, and regions used for epithelial downstream analysis are represented (cohort #2). Scale bars, 50 μm (representative of 1 independent experiment).

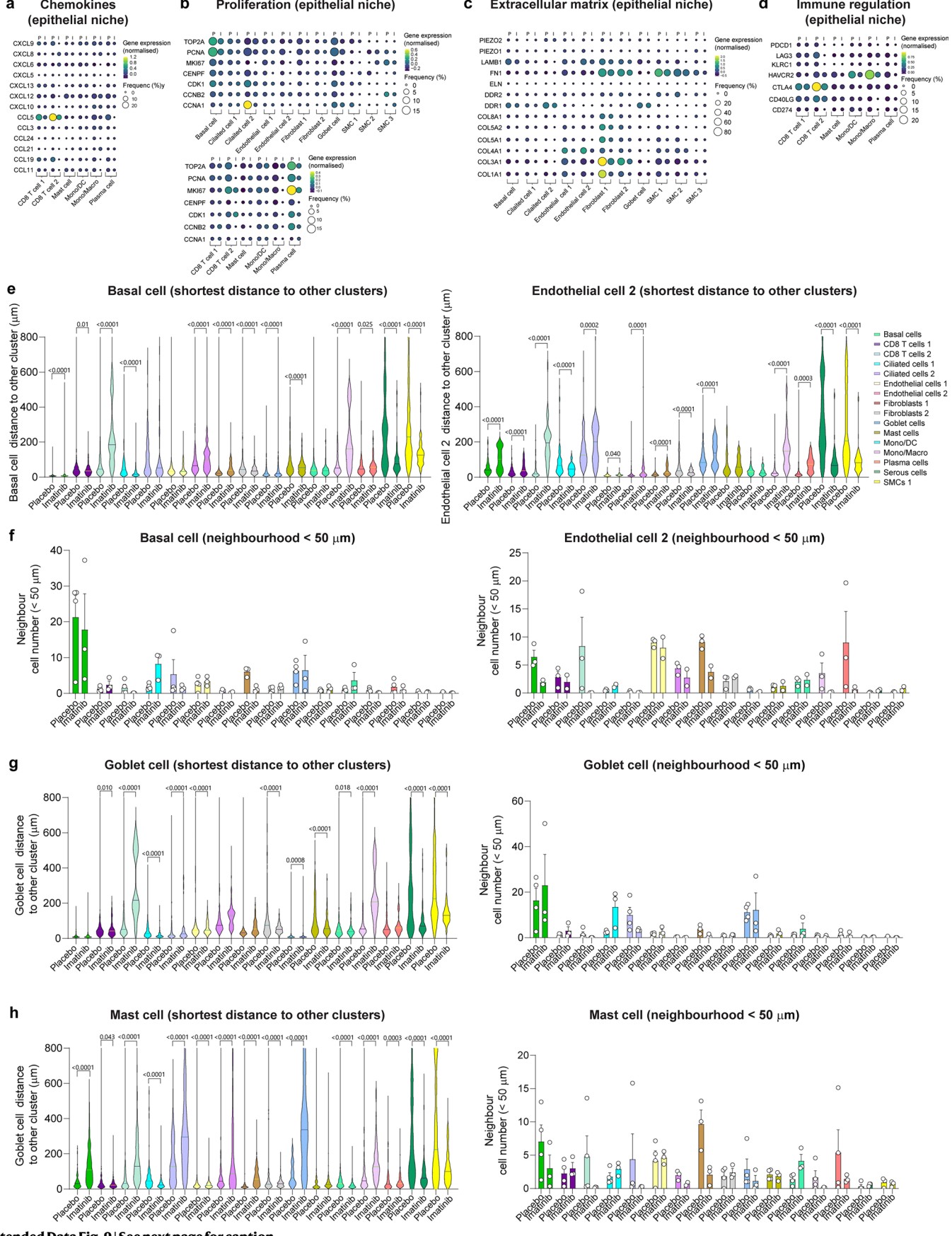

**Extended Data Fig. 9 | See next page for caption.**

**Extended Data Fig. 9 | Epithelial niche of placebo and imatinib treated donors for alarmins, chemokines, proliferation, extracellular matrix, immune regulation and spatial organisation. a**, Gene dot plot for chemokine expression in immune cell clusters (cohort #2). **b**, Gene dot plot for proliferation markers expression in all clusters. **c**, Gene dot plot for extracellular matrix gene expression in stroma cell clusters. **d**, Gene dot plot for immune regulation gene expression in immune cell clusters. **e**, Shortest distance between basal cell and endothelial cell 2 to each cluster in the epithelial niche (pooled cells n = 3/4 donors per group, cohort #2). **f**, Average neighbouring cell number between basal cell and endothelial cell 2 to each cluster in the epithelial niche (n = 4

(placebo) and 3 (imatinib) donors per group, cohort #2). Mean ± SEM. **g**, Shortest distance and average neighbouring cell between goblet cell to each cluster in the epithelial niche (n = 4 (placebo) and 3 (imatinib) donors per group, cohort #2). Mean ± SEM. **h**, Shortest distance and average neighbouring cell between mast cell to each cluster in the epithelial niche (n = 4 (placebo) and 3 (imatinib) donors per group, cohort #2). Mean ± SEM. Columns alternate between placebo (P), and imatinib (I) samples. Scales are provided for each plot, red boxes indicate genes with p value < 0.05 compared to other clusters, black boxes indicate genes with p value < 0.05 between placebo and imatinib. Two-sided Mann-Whitney t test.

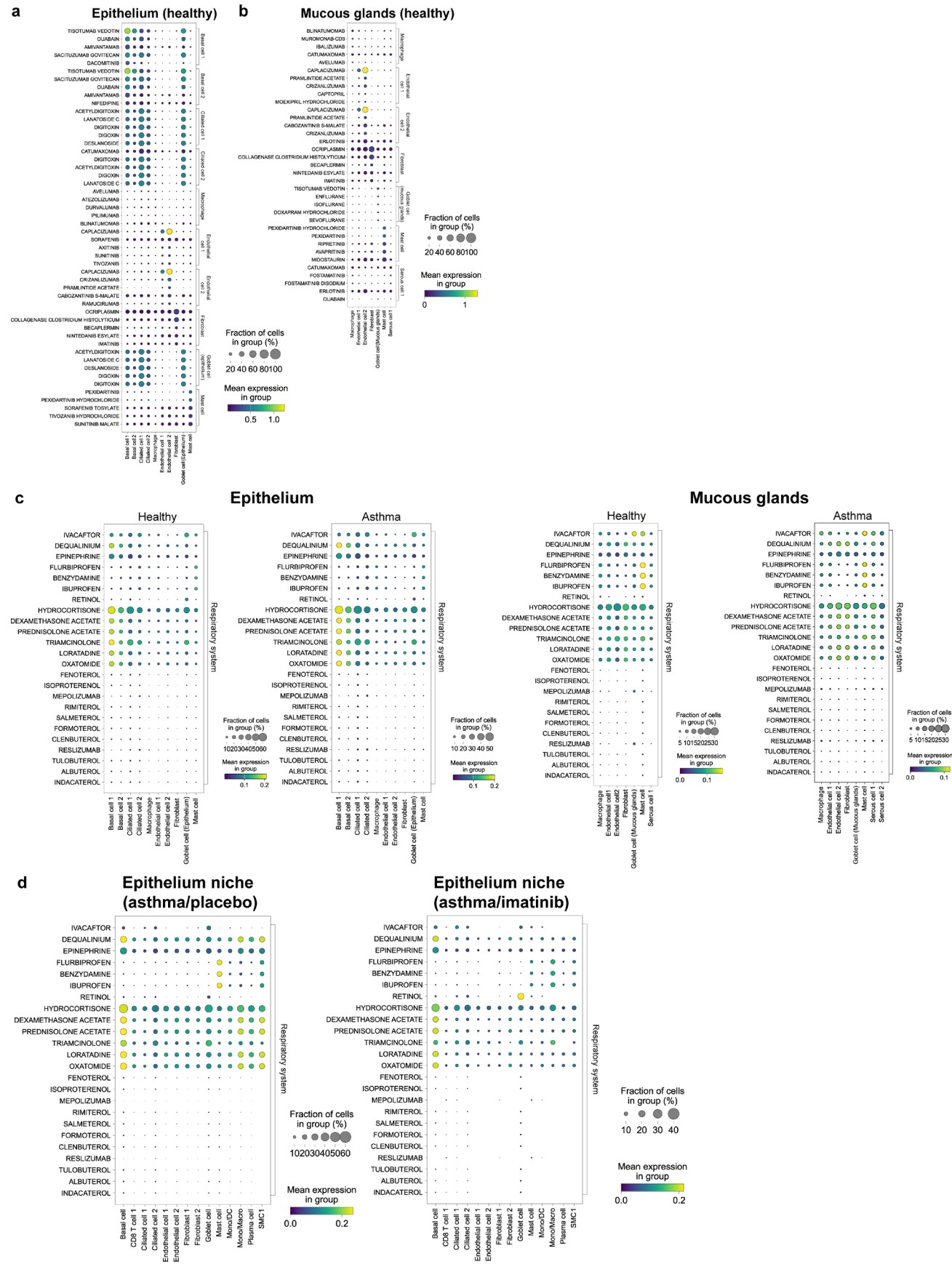

**Extended Data Fig. 10 | See next page for caption.**

**Extended Data Fig. 10 | Spatial drug2cell analysis of lung biopsies in health and asthma with respiratory drugs. a-b**, Top 5 drugs per cluster with strongest interaction and frequency of cells impacted in the epithelium (**a**) and mucous glands (**b**) areas of healthy donors (cohort #1). **c-d**, Respiratory approved drugs interaction with cell clusters in the epithelium or mucous glands areas of healthy and asthma biopsies (**c**, cohort #1) or epithelium of asthma patients treated with placebo and imatinib (**d**, cohort #2). For drug dot plot, size of the dot reflects frequency of cell within the cluster impacted by the drug and colour indicates the interaction strength.

# Reporting Summary

## Statistics

For all statistical analyses, confirm that the following items are present in the figure legend, table legend, main text, or Methods section.

| n/a | Confirmed | |
|---|---|---|
| ☐ | ☒ | The exact sample size (*n*) for each experimental group/condition, given as a discrete number and unit of measurement |
| ☐ | ☒ | A statement on whether measurements were taken from distinct samples or whether the same sample was measured repeatedly |
| ☐ | ☒ | The statistical test(s) used AND whether they are one- or two-sided *Only common tests should be described solely by name; describe more complex techniques in the Methods section.* |
| ☒ | ☐ | A description of all covariates tested |
| ☐ | ☒ | A description of any assumptions or corrections, such as tests of normality and adjustment for multiple comparisons |
| ☐ | ☒ | A full description of the statistical parameters including central tendency (e.g. means) or other basic estimates (e.g. regression coefficient) AND variation (e.g. standard deviation) or associated estimates of uncertainty (e.g. confidence intervals) |
| ☐ | ☒ | For null hypothesis testing, the test statistic (e.g. *F*, *t*, *r*) with confidence intervals, effect sizes, degrees of freedom and *P* value noted *Give P values as exact values whenever suitable.* |
| ☒ | ☐ | For Bayesian analysis, information on the choice of priors and Markov chain Monte Carlo settings |
| ☒ | ☐ | For hierarchical and complex designs, identification of the appropriate level for tests and full reporting of outcomes |
| ☒ | ☐ | Estimates of effect sizes (e.g. Cohen's *d*, Pearson's *r*), indicating how they were calculated |

*Our web collection on statistics for biologists contains articles on many of the points above.*

## Software and code

Policy information about availability of computer code

| Data collection | LEICA LAS (version X), Xenium platform (v3.0), Nanostring DSP (V3.0.0.111) |
|---|---|
| Data analysis | Imaris (v10.1), LEICA LAS version X, R studio (version 2023.06.01), Xenium explorer (version 3.1.1, 10X), Prism software (GraphPad v9.4.1), R (v4.3.0), Python(v3.12), Nanostring DSP analysis suite (GEOMX-B0007), UMAP (0.2.10.0), ggplot2 (3.5.1), sctransform (0.4.1), Seurat (5.2.1), phenoptr (0.3.2), drug2cell (v0.1.0), SquidPy (v1.4.1), GSVA (2.0.1), Stringr (1.5.1), Dplyr (1.1.4) |

For manuscripts utilizing custom algorithms or software that are central to the research but not yet described in published literature, software must be made available to editors and reviewers. We strongly encourage code deposition in a community repository (e.g. GitHub). See the Nature Portfolio guidelines for submitting code & software for further information.

## Data

Policy information about availability of data

All manuscripts must include a data availability statement. This statement should provide the following information, where applicable:
- Accession codes, unique identifiers, or web links for publicly available datasets
- A description of any restrictions on data availability
- For clinical datasets or third party data, please ensure that the statement adheres to our policy

The analysis was performed using published and freely available software and code mentioned in relevant method sections. Raw and processed data are available at

## Research involving human participants, their data, or biological material

Policy information about studies with underline{human participants or human data}. See also policy information about underline{sex, gender (identity/presentation), and sexual orientation} and underline{race, ethnicity and racism}.

| | |
|---|---|
| Reporting on sex and gender | We analysed samples from 17 females and 9 males, this is reflective of the availability of donors and the predispostion of asthma in women. KIA samples information are available from the oringial publication of the clinical trial (ClinicalTrials.gov number, NCT01097694). |
| Reporting on race, ethnicity, or other socially relevant groupings | Tissues from the first cohort are from the UK population, mainly urban location around London and Leicester. The samples from the KIA study were colelcted in North america and ethnics information can be found in the original publication and clinical trial. |
| Population characteristics | Adult patients (18-65 years old) with severe asthma as defined by European Respiratory Society (ERS) and American Thoracic Society (ATS) guidelines were recruited. All had a BMI of 18.5-35 kg/m2 expect one at 66.11, non-smoker or ex-smoker for at least past 12 months, FEV1 of ≥1 litre AND ≥ 60% predicted. For patient stratification between mild and severe asthma, we employed the European Respiratory Society (ERS) and American Thoracic Society (ATS) guidelines. Mild asthma was characterised by infrequent symptoms that can be managed with low-dose medications. In contrast, moderate to severe asthma is defined by persistent, heavy symptoms despite high-dose medications, frequent exacerbations, and often requires additional therapies such as biologics. |
| Recruitment | All lung tissues were obtained after Research Ethics Committee approvals and informed consent from the donor families. |
| Ethics oversight | The collection of endobronchial biopsies was approved by London - Bloomsbury Research Ethics Committee under the approval REC 19/LO/1675. KIA clinical trial was approved by the Institutional Review Board (IRB) or Independent Ethics Committee (IEC) of Harvard MEdical School Human adult samples used in this research project were obtained from the Imperial College Healthcare Tissue Bank (ICHTB). ICHTB is supported by the National Institute for Health Research (NIHR) Biomedical Research Centre based at Imperial College Healthcare NHS Trust and Imperial College London. ICHTB is approved by Wales REC3 to release human material for research (17/WA/0161), and the samples for this project (R22006) were issued from sub-collection reference number ICB_NC_21_017. The views expressed are those of the authors and not necessarily those of the NHS, the NIHR or the Department of Health. The biopsy samples from University Hospitals of Leicester were issued under MTA 2021S-0809-2029 between University of Leicester/UHL and Imperial College London and were approved by the research ethics committee MREC: 08/H0406 and IRAS: 8824. |

Note that full information on the approval of the study protocol must also be provided in the manuscript.

# Field-specific reporting

Please select the one below that is the best fit for your research. If you are not sure, read the appropriate sections before making your selection.

☒ Life sciences ☐ Behavioural & social sciences ☐ Ecological, evolutionary & environmental sciences

For a reference copy of the document with all sections, see underline{nature.com/documents/nr-reporting-summary-flat.pdf}

# Life sciences study design

All studies must disclose on these points even when the disclosure is negative.

| | |
|---|---|
| Sample size | Sample size was determined to provide meaningfull data with this new technology. As nobody reported this type of data before, we estimated that 4 patients per group will give enough power and this aligned with Human Cell Atlas crtieria. |
| Data exclusions | No data were excluded |
| Replication | We replicated our spatial single cell RNAseq experiment in 3 independant cohorts and different technologies with similar results. Human imaging data such as PCLS were performed on separate part of the tissue. Mutiple fields of view (at least 5) were captured for each part of the tissue. All attempts at replication were successful. |
| Randomization | To avoid batch effects, we run simulatenously samples from healthy and asthma donors or palcebo and imatinib samples. Original clinical trial includes randomization of patients receiving placebo or imatinib. |
| Blinding | Investigators were blinded during data collection for Xenium and GeoMX datasets. No blinding was required for imaging lung samples as no separate groups were compared. Samples were analysed as a whole before being analysed considering the pathology status as such reducing any potential biased. Furthermore, we applied multiple unbiased analysis strategy to discover new elements of lung biology. |

# Reporting for specific materials, systems and methods

We require information from authors about some types of materials, experimental systems and methods used in many studies. Here, indicate whether each material, system or method listed is relevant to your study. If you are not sure if a list item applies to your research, read the appropriate section before selecting a response.

## Materials & experimental systems

| n/a | Involved in the study |
|-----|----------------------|
| ☐ | ☒ Antibodies |
| ☒ | ☐ Eukaryotic cell lines |
| ☒ | ☐ Palaeontology and archaeology |
| ☒ | ☐ Animals and other organisms |
| ☒ | ☐ Clinical data |
| ☒ | ☐ Dual use research of concern |
| ☒ | ☐ Plants |

## Methods

| n/a | Involved in the study |
|-----|----------------------|
| ☒ | ☐ ChIP-seq |
| ☒ | ☐ Flow cytometry |
| ☒ | ☐ MRI-based neuroimaging |

## Antibodies

| | |
|--|--|
| Antibodies used | Alexa Fluor 488 anti-human CD31 (clone WM59, dilution 5 ug/ml) from Biolegend, avidin (#A887) was conjugated to Alexa fluor 647 (#A20186) dilution 3.3 ug/ml, ThermoFisher. |
| Validation | Validation of all primary commercial antibodies for the species and application was warranted by the vendors. Validation statement can be found on the manufacturers' website. For Alexa Fluor 488 anti-human CD31, each lot of this antibody is quality control tested by immunofluorescent staining with flow cytometric analysis. Clone WM59 has been reported to recognize the D2 extracellular portion of CD31. For avidin home made conjugate, in house validation was performed using primary mast cell expressing strongly the recognsied antigen (i.e. heparin). |

## Plants

| | |
|--|--|
| Seed stocks | *Report on the source of all seed stocks or other plant material used. If applicable, state the seed stock centre and catalogue number. If plant specimens were collected from the field, describe the collection location, date and sampling procedures.* |
| Novel plant genotypes | *Describe the methods by which all novel plant genotypes were produced. This includes those generated by transgenic approaches, gene editing, chemical/radiation-based mutagenesis and hybridization. For transgenic lines, describe the transformation method, the number of independent lines analyzed and the generation upon which experiments were performed. For gene-edited lines, describe the editor used, the endogenous sequence targeted for editing, the targeting guide RNA sequence (if applicable) and how the editor was applied.* |
| Authentication | *Describe any authentication procedures for each seed stock used or novel genotype generated. Describe any experiments used to assess the effect of a mutation and, where applicable, how potential secondary effects (e.g. second site T-DNA insertions, mosiacism, off-target gene editing) were examined.* |

