## [Peer Review File · Nature Immunology]

A single-cell spatial chart of the airway wall reveals proinflammatory cellular ecosystems and their interactions in health and asthma

Corresponding Author: Dr Régis Joulia

Version 0:

Decision Letter:

16th Oct 2024

Dear Dr. Joulia,

We have now finished reviewing your manuscript entitled "Single-cell spatial chart of the airway wall reveals novel cellular ecosystems and their interactions in health and asthma", reference number NI-A38560-T.

Although the editors thought that the manuscript was interesting enough to send out for in-depth review, the reviewers were not in favor of publishing the paper in Nature Immunology because of the strength of the novel conclusions that can be drawn at this stage. We are therefore returning the reviews to you with the hope that you find them useful when you prepare the paper for another journal.

Although we cannot publish your paper, it may be appropriate for another journal in the Nature Portfolio. If you wish to explore the journals and transfer your manuscript please use our manuscript transfer portal. You will not have to re-supply manuscript metadata and files, unless you wish to make modifications. For more information, please see our http://www.nature.com/authors/author_resources/transfer_manuscripts.html?WT.mc_id=EMI_NPG_1511_AUTHORTRANSF&WT.ec_id=AUTHOR manuscript transfer FAQ page.

We realize that this is disappointing. I hope that you continue to consider Nature Immunology for your results most significant for the immunology community and wish you well in your future investigations.

Sincerely,

Stephanie Houston, PhD
Senior Editor
Nature Immunology

Reviewers' comments:

Reviewer #1 (Remarks to the Author):

This manuscript by Joulia, Lloyd, and colleagues presents a spatial transcriptomic analysis of lung biopsies from asthma patients and healthy controls along with asthma patients treated with a drug targeting mast cells and other targets. They identify cellular neighborhoods that exhibit either distinct compositions in asthma versus healthy controls, as well as transcriptomic differences between specific lung cell types in asthma versus control biopsies. The strength of the study is in applying spatial transcriptomics analysis to these interesting and hard-to-obtain human samples in a disease context and treated with different drugs. However, there are a number of different aspects of the study that are not well described nor clearly presented which raise questions as to the statistical significance of the results, whether they are powered to detect statistical differences, and how to interpret differences in cellular neighborhoods that are more subtle or due to one cell type. Specific points are enumerated below.

1. The initial spatial dataset appears to be derived from biopsies from 4 asthma samples and 4 controls, but it was difficult to even find out these numbers and whether replicate biopsy samples were obtained. The entire dataset as presented in

Figures 1-3 combines all of these samples, but it the numbers of subjects or replicate samples that these samples derive from was not mentioned in the legends, results, or methods. The information for the subjects in Supplemental Table 1 shows the sample ID, etc. for the two xenium slides used which is a confusing way to present the data and for the second Xenium slide, there appeared to be replicates, but not clear what all of the id and visit and other info provided in the table represents and moreover, what was included in the Xenium dataset. This straightforward information about number of subjects, samples and what the xenium data derived from should be clearly explained for every figure.

2. The statistical significance of the gene expression differences identified between asthma and healthy donors for the different cells (e.g., basal cell types, endothelial cells, mast cells, etc.) in Figs. 1-3 is not mentioned. The dot plots refer to the 20 top expressed genes for a given cell population but how are the differences in gene expression between healthy and asthma samples assessed? Is this done by paired analysis—for each donor or sample? The same question applies to the samples obtained from the treated individuals in Figs. 4 and 5. There is no mention of significance cut-offs or graphs showing the expression in the individual subjects using heat maps or other such visualizations.

3. For the neighborhood analysis, the pie chart visualization shows few differences between asthma and healthy donors for the different neighborhoods, aside from a few differences in c=some cell populations. How were these differences assessed for significance? Did the neighborhood analysis include randomization controls? Were different areas of the biopsies assessed?

4. In all of the figures, the labeling on the images and dot plots uses very tiny fonts, such that e even magnifying the figures 1.5-2-fold, they were still difficult to read. I realize these are complex datasets to present, but the authors could consider focusing on fewer aspects or those results that showed significance between groups and visualizing them more clearly.

Reviewer #2 (Remarks to the Author):

Recent studies have revealed the cellular complexity in lung development and disease, particularly altered cellular profiles in asthma, but often miss crucial spatial information about cell interactions due to isolation methods. The authors used the Xenium platform to analyze endobronchial biopsies from healthy and severe asthma donors undergoing anti-inflammatory treatments to determine the spatial transcriptional signature of the lung bronchial airway. They detected more than 2 million transcripts in ~60,000 cells. The authors utilized a 339 gene panel to identify cell lineages including epithelial cells, endothelial cells and immune cells and optimized section placements to run biopsies from healthy donors and donors with asthma simultaneously. Employing single-cell spatial transcriptomics, the authors explored the transcriptional and spatial dynamics of the bronchial wall, revealing distinct cellular neighborhoods and interactions, including altered mast cell function. The authors demonstrated the impact of the KIT inhibitor imatinib on the asthma microenvironment after 24 weeks, offering insights into drug interactions within the lung tissue. There are some suggestions and comments that may improve the manuscript. As the discussion of the manuscript is weak, I suggested some references for some of the comments. The authors may consider them to improve their manuscript.

1. The author must check and discuss if the findings align with previous studies in the Human Lung Atlas (PMID: 37291214; PMID: 33208946; PMID: 38830933; PMID: 37308579; PMID: 38253462; PMID: 39232171).

2. How should the relevant parameters for quality control be determined, considering that there is currently no consensus on this issue? (<https://www.biorxiv.org/content/10.1101/2024.04.03.586404v1.full.pdf>; PMID: 37353093). Were the obtained results validated using another method?

3. Did the authors detect neuroendocrine cells in their system ?

4. The role of the epithelial barrier in the development of allergic and autoimmune diseases is well-established. Mast cells, through their release of inflammatory mediators, can compromise the integrity of the epithelial barrier, exacerbating disease. Please discuss the effects of mast cells on the bronchial epithelial barrier and how the use of Imatinib, specifically through its inhibition of mast cells, impacts the bronchial epithelial barrier and the subsequent effects on disease progression

5. L76. "The authors detected more than 2 million transcripts in ~60,000 cells and unbiased clustering revealed 18 different clusters with minor differences across healthy and asthma donors." Please explain these minor differences.

6. L91. "phenopt pipeline developed by Akoya Bioscience" Please add a reference.

7. L97. "were located within 25 μ m range, indicating the existence of specific neighbourhoods" Please explain and support this with references because cell density and size varies in the lung with 40 discrete cell types such as ciliated cells; goblet cells; basal cells; brush cells; neuroendocrine cells; stem cells etc. (PMID: 37291214; PMID: 30995076)

8. L107. The authors merge some sections, but it needs more explanation because the manuscript is based on these.

9. L110-113. For figure 2b. Using cNMF and XGBoost, we classified malignant NSCLC cells into three functional modules, identified key marker genes related to tumor evolution (The expression of genes such as CHCHD2, GAPDH, and CD24 was strongly correlated with the malignant), and validated a risk score model with GEO data. Which cells play pathological role in asthma? Please explain clearly (PMID: 38872167)

10. L113. "for IL-33 and chemokines" Please explain clearly

11. L120 "Asthma patients showed a global reduction in all chemokines and alarmins analysed" Some researchers showed that IL-33 upregulated in asthma Please clarify/double check and discuss it (PMID: 38821053; PMID: 38597952; PMID: 34402091; PMID: 34928757)

12. L120-123: It is essential to detail the specific medications used by the patients, including their dosages and duration of treatment, to establish a correlation with the observed data. Analyzing these relationships can provide insights into how treatment protocols contribute to the changes in biomarkers observed in the study. Please explain it.

13. Are all asthma patients (endotypes) from whom biopsy samples were taken homogeneously, and was statistical analysis conducted considering factors such as drug resistance, race, gender, age, and specific exposure? Please explain it.

14. How was the impact of receptor-ligand interactions in spatial interactions between cells, as well as endocrine and paracrine relationships, evaluated in the study?
15. L139-141. Please compare with non-asthmatic samples and explain 'within close distance' quantitatively.
16. L155-157. "Mast cells were abundant within the lung biopsies" Please re-write it and check.
17. L166. "VEGFR2 and IL-7R that are vital for IL-33, vascular" Please clarify if this is the finding of the current manuscript.
18. L184. (our annotated mucous gland areas)The exact location from which the biopsy samples are taken is important because there are differences in cell types and mucus density in the sections of the lung parenchyma and bronchi. Please discuss it.
19. L240. The authors did not above provide information about tryptase in the upper sections of the text, but the authors did not conduct any measurements, or they did not mention or emphasize this in the manuscript.

Reviewer #3 (Remarks to the Author):

General Comments

The authors use spatial transcriptomics technology to examine gene expression in situ in endobronchial biopsy sections from patients with asthma and controls. The application of this technology in this way is powerful, but the paper reads more like a feasibility result for the utility of the technology rather than one that uncovers new findings. In particular, the abstract text is overly committed to emphasizing the utility of the spatial transcriptomic technology at the expense of providing more granular detail about the main study findings. Other concerns include insufficient detail about the human subjects numbers, the collection and quality assurance of biospecimen collection and quality control, and the rigor of statistical analysis.

Specific Comments

1. Human subjects.

- the supplemental table 1 is a densely populated excel spreadsheet, not the summary table that readers need. A table that summarizes the clinical and demographic features of the asthma patients and controls should be provided.
- the main manuscript needs to indicate how many asthma subjects and controls were included in the study.

2. Biospecimens;

- there is little data on the biospecimens and how they were collected. How were the biospecimens obtained? How many biopsies were collected per subjects? Were any quality measures used to classify biopsies as acceptable or not?
- endobronchial biopsies frequently have epithelial cell denudation because of processing artefact. How was this issue handed?
- submucosal gland are variably present in endobronchial biopsies – was any methods feature included to determine when glands could be analyzed?

3. Imatinib trial biospecimens and details

- details are needed for how many human subjects were included and how many days or weeks were patients treated for. What drug dose was used?

4. Mast cells data

- the number of mast cells is very low on the UMAP plots (Figure 2 and Figure 3). What is the actual number of these cells? Can authors provide more information about cell numbers in total and in the different samples?
- mast cells are not shown in the images in Figures 2 and 3, so the "clear abundance of mast cells in proximity to the most pro-inflammatory areas" is hard to evaluate and visualize.

5. Mucin gene data

- usually patients with asthma show an increase in gene expression for MUC5AC and a decrease in MUC5B. The authors find different results in their analyses for MUC5B. Was MUC5AC included in the panel? Can the authors comment in more detail on this observation and refer to the literature?

6. Basal cell clusters

- how do the authors justify that the 2 basal cell clusters cannot be found in all datasets? Are they biologically relevant? Have they been described in other studies?

7. Chemokines and alarmins

- the decrease in chemokines and alarmins in asthma is surprising. Was the number of transcripts per cell the same between cohorts?

8. Data presentation and interpretation

- percentages should be added to the graphs showing the distribution of cellular neighborhoods because the increase or decrease in interaction between cell types is not always clear. There is currently very little visualization of sample distribution.
- remodeling signature; that authors state that: "However, we observed strong tissue remodeling signatures with reduced distances between inflammatory cells and stromal cells" (lines 179-180). What exactly is the remodeling signature? Can authors bring more information on how they jump to this conclusion ?
- the authors discuss pro-inflammatory ecosystems, but it seems that these ecosystems are only present in healthy samples, as shown by the decrease expression of chemokines and alarmins in asthma samples.
- when the authors discuss increases or decreases in gene expression or spatial interaction, these differences are not

justified by statistical tests and seem to refer to simple observations.

Version 1:

Decision Letter:

27th Feb 2025

Dear Dr Joulia,

Your Article, "A single-cell spatial chart of the airway wall reveals proinflammatory cellular ecosystems and their interactions in health and asthma" has now been seen by 3 referees. You will see from their comments below that while they find your work of interest, some important points are raised. We are very interested in the possibility of publishing your study in Nature Immunology, but would like to consider your response to these concerns in the form of a revised manuscript before we make a final decision on publication. We anticipate that remaining concerns can be addressed by making changes to the text. Furthermore, we ask you to resubmit your manuscript in the format of a Resource rather than an Article.

We therefore invite you to revise your manuscript taking into account all reviewer and editor comments. Please highlight all changes in the manuscript text file in Microsoft Word format.

* If you have not done so already please begin to revise your manuscript so that it conforms to our Article format instructions at <http://www.nature.com/ni/authors/index.html>. Refer also to any guidelines provided in this letter.

* Please include a revised version of any required reporting checklist. It will be available to referees to aid in their evaluation of the manuscript goes back for peer review. They are available here:

Reporting summary:

Please note, Extended Data figures and tables are online-only (appearing in the online PDF and full-text HTML version of the paper), peer-reviewed display items that provide essential background to the Article but are not included in the printed version of the paper due to space constraints or being of interest only to a few specialists. A maximum of ten Extended Data display items (figures and tables) is typically permitted. When re-submitting your manuscript, please ensure that any supplementary figures and tables that are more critical to the manuscript's conclusions are converted to Extended data to increase these data's visibility.

Link Redacted

We hope to receive your revised manuscript within two weeks. If you cannot send it within this time, please let us know. We will be happy to consider your revision so long as nothing similar has been accepted for publication at Nature Immunology or published elsewhere.

Nature Immunology is committed to improving transparency in authorship. As part of our efforts in this direction, we are now requesting that all authors identified as 'corresponding author' on published papers create and link their Open Researcher and Contributor Identifier (ORCID) with their account on the Manuscript Tracking System (MTS), prior to acceptance. ORCID helps the scientific community achieve unambiguous attribution of all scholarly contributions. You can create and link your ORCID from the home page of the MTS by clicking on 'Modify my Springer Nature account'. For more information please visit www.springernature.com/orcid.

Sincerely,

Stephanie Houston, PhD
Senior Editor
Nature Immunology

Reviewers' Comments:

Reviewer #1 (Remarks to the Author):

The authors have addressed my major concerns regarding statistical analysis and improved the presentation of the figures such that they are much clearer now.

Reviewer #2 (Remarks to the Author):

Thank you for a full revision of the manuscript.

Reviewer #3 (Remarks to the Author):

The authors provide a revised manuscript with additional data from 10 more patients with mild/severe asthma and 5 healthy controls and there is improved statistical analyses and rigor. I have the following comments:

1. It is not clear that the main message of the paper advances the field. The authors state that their main finding is that the airway wall in health and in asthma has discrete cellular areas ("cellular ecosystems") that are enriched in cells producing alarmins and chemokines. Surprisingly, these cellular ecosystems were more prominent in healthy controls than in asthma prompting the authors to speculate that they protect against pathogens and are dysregulated during asthma. But this is not convincing because their data show that the alarmins that are thought to drive type 2 inflammation in particular are decreased in asthma biopsies. It is possible that there is a negative feedback or some other explanation for this finding but the lack of clarity is a weakness.

2. The data for cell characterization within the ecosystems or hubs is presented in a confusing manner and the figures, legends and manuscript text remain hard to read through and fully understand. There is a strong reliance in the paper on general statements about spatial mapping and landscape description for its own sake.

3. Although additional human characterization data is provided in Supplementary tables 1 and 2, the information provided is hard to assimilate because data about inhaled medications generally missing and the lung function data is inconsistently presented (FEV1 data for some and FEV1/FVC ratio for others. It is not clear how patients were subdivided into mild vs severe asthmatics, a subgrouping which is used in some of the figures.

Version 2:

Decision Letter:

Our ref: NI-RS38560B

13th Mar 2025

Dear Dr. Joulia,

Thank you for submitting your revised manuscript "A single-cell spatial chart of the airway wall reveals proinflammatory cellular ecosystems and their interactions in health and asthma" (NI-RS38560B). We'll be happy in principle to publish it in Nature Immunology, pending minor revisions to comply with our editorial and formatting guidelines.

We will now perform detailed checks on your paper and will send you a checklist detailing our editorial and formatting requirements in about a week. Please do not upload the final materials and make any revisions until you receive this additional information from us.

If you had not uploaded a Word file for the current version of the manuscript, we will need one before beginning the editing process; please email that to immunology@us.nature.com at your earliest convenience.

Thank you again for your interest in Nature Immunology Please do not hesitate to contact me if you have any questions.

Sincerely,

Stephanie Houston, PhD
Senior Editor
Nature Immunology

Point by point response to Comments on NI-A38560-T

We would like to sincerely thank the editor and all three reviewers for their comments, which we found insightful and constructive. We are now confident that we have addressed all the points raised, as outlined in the attached document.

The modified text in the manuscript is highlighted in red.

Main points addressed:

1. Lack of detail on patients and clinical samples included: We have generated a simplified table detailing the patients and controls, their demographics and treatment regimen (Supplemental table 1).

2. Strength of Novel Conclusions/Number of Subjects Analysed: Single-cell spatial transcriptomics is a groundbreaking technology with immense potential for both fundamental research and clinical application. As reviewer #2 correctly pointed out, while recent studies have unveiled the cellular complexity of the lungs, they lacked spatial information. Our study is the first to provide a comprehensive spatial map of the lung in both health and in the context of a severe immune disorder: asthma. We believe our conclusions are both strong and novel, particularly the identification of specialised populations of basal and endothelial cells, along with their unique profiles and their capacity to secrete alarmins and chemokines.

We have analysed **12 patients with asthma and 4 healthy controls** using single-cell spatial transcriptomics with one of the largest relevant panel of genes. To our knowledge, this level of analysis has not been previously achieved. However, to further strengthen the manuscript and assuage the reviewers' concerns, we included **additional data from 10 more patients with mild/severe asthma and 5 healthy controls** using a complementary spatial transcriptomics platform (GEOMx, Nanostring), with **a full transcriptome analysis** generated from an entirely different asthma/control cohort. As shown from the new Figure, this approach integrates single-cell spatial data with full transcriptomic profiles from human tissue sections in both health and disease:

In summary, our study encompass data from over **22 asthma patients and 9 healthy controls**. This also significantly enhances the robustness of our scientific conclusions and addresses the statistical concerns raised by the reviewers as detailed in the point by point detailed below.

3. General clarity of the Manuscript: The original submission was formatted for *Nature*, and we believe the shorter format (including small figures) may have contributed to some of the confusion among the reviewers. In our revised manuscript, we have expanded and clarified key sections to address the comments from all three reviewers and ensure our findings are presented with greater clarity and detail.

Reviewer #1 (Remarks to the Author):

This manuscript by Joulia, Lloyd, and colleagues presents a spatial transcriptomic analysis of lung biopsies from asthma patients and healthy controls along with asthma patients treated with a drug targeting mast cells and other targets. They identify cellular neighborhoods that exhibit either distinct compositions in asthma versus healthy controls, as well as transcriptomic differences between specific lung cell types in asthma versus control biopsies. The strength of the study is in applying spatial transcriptomics analysis to these interesting and hard-to-obtain human samples in a disease context and treated with different drugs. However, there are a number of different aspects of the study that are not well described nor clearly presented which raise questions as to the statistical significance

of the results, whether they are powered to detect statistical differences, and how to interpret differences in cellular neighborhoods that are more subtle or due to one cell type. Specific points are enumerated below.

We would like to thank the reviewer for their insightful comments and we agree on the challenge of obtaining and analysing human lung samples from patients with asthma. We have now provided a revised version of the manuscript incorporating all their comments and we believe to have answers all the point raised.

1. The initial spatial dataset appears to be derived from biopsies from 4 asthma samples and 4 controls, but it was difficult to even find out these numbers and whether replicate biopsy samples were obtained. The entire dataset as presented in Figures 1-3 combines all of these samples, but it the numbers of subjects or replicate samples that these samples derive from was not mentioned in the legends, results, or methods. The information for the subjects in Supplemental Table 1 shows the sample ID, etc. for the two xenium slides used which is a confusing way to present the data and for the second Xenium slide, there appeared to be replicates, but not clear what all of the id and visit and other info provided in the table represents and moreover, what was included in the Xenium dataset. This straightforward information about number of subjects, samples and what the xenium data derived from should be clearly explained for every figure.

We thank the reviewer for their helpful comments that allowed us to clarify numerous aspects of our study. Our initial study included samples from 4 donors with asthma and 4 healthy samples (one biopsy per patient) and 8 donors with asthma treated with either placebo or imatinib. Before performing single cell spatial transcriptomics, we carefully checked that all our samples validated our strict characteristics for analysis such as good architecture as visible by multilayer epithelium, presence of inflammatory cells and correct smooth muscle distribution.

Although this could perhaps be conceived as limited, the number of subjects analysed is in line with original studies published in the field. *Travaglini et al. Nature 2020*, employed 3 donors for their lung cell atlas. *Vieira Braga et al. Nature Medicine 2019*, analysed 36,931 single cells from 6 healthy and 7 asthma donors. *Deprez et al. Am J Resp Crit Care Med 2020*, analysed 77,969 cells from 10 donors. *Madissoon et al. Nature genetics 2022*, analysed 193,108 cells from 11 deceased donors. Our study reached a total 106,304 cells analysed from 15 donors, a significant improvement from previous studies.

Furthermore, we have confirmed our data using an independent cohort and different platform to reach a total of **8 healthy and 20 asthma donors**, thus representing an unprecedented progression in the lung field.

2. The statistical significance of the gene expression differences identified between asthma and healthy donors for the different cells (e.g., basal cell types, endothelial cells, mast cells, etc.) in Figs. 1-3 is not mentioned. The dot plots refer to the 20 top expressed genes for a given cell population but how are the differences in gene expression between healthy and asthma samples assessed? Is this done by paired analysis—for each donor or sample? The same question applies to the samples obtained from the treated individuals in Figs. 4 and 5. There is no mention of significance cut-offs or graphs showing the expression in the individual subjects using heat maps or other such visualizations.

We agree with the reviewer that our original statistical visualisation was not optimal, and we have significantly improved this aspect. As shown in the revised manuscript, genes significantly different (p value <0.05) for a specific cluster are indicated by a red box and genes significantly different between healthy and asthma or placebo and imatinib are shown by a black box. We performed an unpaired analysis between healthy and asthma samples considering all the cells together as gene expression has been normalised allowing for direct comparison.

We added details for individual subject spatial analysis as shown in the revised Fig. 3, 5 and Extended data Fig. 2, 4, 5, 6, 7, 9.

3. For the neighborhood analysis, the pie chart visualization shows few differences between asthma and healthy donors for the different neighborhoods, aside from a few differences in c=some cell populations. How were these differences assessed for significance? Did the neighborhood analysis include randomization controls? Were different areas of the biopsies assessed?

The reviewer is perfectly right that the pie charts were not ideal to show spatial organisation. We updated our manuscript with new visualisations showing repartition of neighbours from 15 to 150 μm and 2 additional statistical analyses. First, we showed for each cluster of cells their shortest distance to other clusters and secondly the number of cells surrounding them within specific distance. This series of analyses showed that during asthma, whilst we did not observe an increase in the number of neighbouring cells, we showed that they are clustering together as shown by statistical difference in the shortest distance. We did not include randomization control as we do not think this model applies to the stratified structure of the lung epithelium. Whole biopsy was assessed when the structure of the tissue was intact as shown in Fig. 2 and Extended Data Fig. 2 where we showed the areas analysed. Furthermore, as these biopsies presented different lung structures (i.e. epithelium and submucosal glands), we separated our analysis in two parts and characterised specialised mechanisms in both niches.

4. In all of the figures, the labeling on the images and dot plots uses very tiny fonts, such that e even magnifying the figures 1.5-2-fold, they were still difficult to read. I realize these are complex datasets to present, but the authors could consider focusing on fewer aspects or those results that showed significance between groups and visualizing them more clearly.

We apologise for the small labelling, and we have updated all our main and supplementary figures to increased font size and clarity of the figures.

Reviewer #2 (Remarks to the Author):

Recent studies have revealed the cellular complexity in lung development and disease, particularly altered cellular profiles in asthma, but often miss crucial spatial information about cell interactions due to isolation methods. The authors used the Xenium platform to analyze endobronchial biopsies from healthy and severe asthma donors undergoing anti-inflammatory treatments to determine the spatial transcriptional signature of the lung bronchial airway. They detected more than 2 million transcripts in ~60,000 cells. The authors utilized a 339 gene panel to identify cell lineages including epithelial cells, endothelial cells and immune cells and optimized section placements to run biopsies from healthy donors and donors with asthma simultaneously. Employing single-cell spatial transcriptomics, the authors explored the transcriptional and spatial dynamics of the bronchial wall, revealing distinct cellular neighborhoods and interactions, including altered mast cell function. The authors demonstrated the impact of the KIT inhibitor imatinib on the asthma microenvironment after 24 weeks, offering insights into drug interactions within the lung tissue. There are some suggestions and comments that may improve the manuscript. As the discussion of the manuscript is weak, I suggested some references for some of the comments. The authors may consider them to improve their manuscript.

We thank the reviewer for their comments on our manuscript and have now implemented the changes suggested.

1. The author must check and discuss if the findings align with previous studies in the Human Lung Atlas (PMID: 37291214; PMID: 33208946; PMID: 38830933; PMID: 37308579; PMID: 38253462; PMID: 39232171).

This is critical point raised by the reviewer and we would like to thank them for raising this point. We have indeed used the Human Lung Atlas extensively to compare our data with their datasets. As shown in Extended data Fig. 1 and 7, we projected all our cell cluster signatures and confirmed our cell annotation and the presence of our cells in other datasets.

Most samples used from the lung cell atlas come from deceased patients, whereby tissue is digested to generate a single cell suspension and are very large encompassing multiple regions of the lung beyond the specific lung bronchial niches analysed here. Although, we believe this resource is incredibly helpful, one needs to be careful when comparing data as they may not represent the specific cellular diversity of the bronchial wall. We have now added these elements in our revised discussion and added the references suggested.

2. How should the relevant parameters for quality control be determined, considering that there is currently no consensus on this issue? (<https://www.biorxiv.org/content/10.1101/2024.04.03.586404v1.full.pdf>; PMID: 37353093). Were the obtained results validated using another method?

We fully agree with the reviewer comment that there is currently no consensus on quality control for spatial transcriptomics. Combined together, the authors have decades of experience in analysing tissue architecture in endobronchial biopsies and are confident that all the tissues analysed here are a good reflection of asthma pathology. Furthermore, as suggested by the reviewer, we confirmed our data using an independent cohort of patients and separated technology (Nanostring GeoMX). We showed similar distribution of chemokines, reduction of IL-33, increased MUC5B and APOD.

3. Did the authors detect neuroendocrine cells in their system ?

We included 7 cardinal genes for neuroendocrine cells in our panels (ASCL1, ICA1, DIRAS3, MAP7, RBP4, SEC11C, TM4SF4). However, we could not detect sufficient expression of these genes to analyse their profiles using Xenium (Fig. 1). This may reflect the rarity of these cells in this specific area of the human lung.

Figure 1. Expression of neuroendocrine cell genes in Xenium samples. Representative Xenium picture of neuroendocrine cell gene expression (yellow dots) and transcript intensity is reflected by the false colour intensity map.

4. The role of the epithelial barrier in the development of allergic and autoimmune diseases is well-established. Mast cells, through their release of inflammatory mediators, can compromise the integrity of the epithelial barrier, exacerbating disease. Please discuss the effects of mast cells on the bronchial epithelial barrier and how the use of Imatinib, specifically through its inhibition of mast cells, impacts the bronchial epithelial barrier and the subsequent effects on disease progression

Mast cells and their mediators have indeed been shown to disrupt barrier integrity. Specifically, we have shown that mast cell tryptase was able to destabilise blood vessel integrity by damaging cell/cell interaction between endothelial cells and pericytes (*Jouliia et al. JCI 2024*). Additional studies have shown that mast cell proteases disrupt epithelial cell junctions and morphology (*Berlin et al. Int J Mol*

Sci 2021, Ramu et al. BMC Immunology 2021, Pejler ERS journal 2019). As the reviewer pointed out, imatinib had a profound impact on mast cell mediators and the level of CPA3, a major mast cell protease, was greatly reduced. This could have an impact on restoring the barrier integrity. We thank the reviewer for highlighting these ideas and we have added these interesting elements added to the discussion.

5. L76. "The authors detected more than 2 million transcripts in ~60,000 cells and unbiased clustering revealed 18 different clusters with minor differences across healthy and asthma donors." Please explain these minor differences.

These minor differences at the global level are related to the difference in serous cells presented in the new Figure 4. We characterised that serous cells are so altered during asthma that they appear as a different cluster expressing high level of lactoferrin and mucous. The text in the result section has been clarified to explain these differences.

6. L91. "phenoptr pipeline developed by Akoya Bioscience" Please add a reference.

We have inserted the reference both in the method section and results.

"Johnson, K. J. R. p. v. phenoptr: inForm helper functions. 9 (2020)"

7. L97. "were located within 25 μm range, indicating the existence of specific neighbourhoods" Please explain and support this with references because cell density and size varies in the lung with 40 discrete cell types such as ciliated cells; goblet cells; basal cells; brush cells; neuroendocrine cells; stem cells etc. (PMID: 37291214; PMID: 30995076)

We thank the reviewer for highlighting this. We have now improved our representation of spatial organisation to clarify this important point. In the new version of the manuscript, we present data analysing repartition of neighbours from 15 to 150 μm . This allowed us to encompass different cell shapes as well as their organisation. Thanks to this method of visualisation, we have characterised along this specific distance (15 to 150 μm) how tissue cellular organisation is disrupted and showed that this was due to specific cell types were aggregating around each other.

8. L107. The authors merge some sections, but it needs more explanation because the manuscript is based on these.

Indeed, we classified our biopsies into two areas: epithelial and mucous gland area. As shown recently in the spatially resolved atlas of the healthy human lung (*Madisson et al. Nature Genet 2023*), the human airway wall has an abundance of submucous glands important for immune response and tissue regeneration with a unique transcriptional signature. Similarly, we showed that our transcripts were segregated into two areas and we believe that spatial interaction within these two different niches are critical for the function of the bronchial wall. We better highlighted this aspect in the revised version of our manuscript.

9. L110-113. For figure 2b. Using cNMF and XGBoost, we classified malignant NSCLC cells into three functional modules, identified key marker genes related to tumor evolution (The expression of genes such as CHCHD2, GAPDH, and CD24 was strongly correlated with the malignant), and validated a risk score model with GEO data. Which cells play pathological role in asthma? Please explain clearly (PMID: 38872167)

We would like to thank the reviewer to point out this very interesting paper. Unfortunately, we did not have CHCHD2 and GAPDH in our panel and CD24 was only weakly expressed (Fig. 2) - this may reflect the very different pathology of NSCLC and asthma. Thus, we could not analyse if asthma was also characterized by high expression of these genes. We would be happy to reference this paper if the editor and reviewer feel that it is important

Fig. 2: Expression of CD24 in Xenium. Representative picture of asthma biopsy using Xenium showing CD24 expression (red dot) and false colour intensity to show transcript enrichment.

Many cells play a pathological role in asthma and although various studies identified target, our study is the first to show them in a spatial context in the human lung, to determine how they interact with each other.

10. L113. “for IL-33 and chemokines” Please explain clearly

We and other have shown that IL-33 and chemokines play a key role during asthma by mediating the recruitment and activation of immune cells (*Puttur et al. Sci Immunol 2019, Saglani et al. Sci Immunol 2018, Andersson et al. JACI 2017, Castanhinha et al. JACI 2015, Saglani et al. JACI 2013*). As these mediators are highly regulated at the genetic level, we believe that their spatial regulation is an essential part of asthma pathology, and we shed new light on the expression of these mediators and how their location differs in our samples.

11. L120 “Asthma patients showed a global reduction in all chemokines and alarmins analysed” Some researchers showed that IL-33 upregulated in asthma Please clarify/double check and discuss it (PMID: 38821053; PMID: 38597952; PMID: 34402091; PMID: 34928757)

We would like to thank the reviewer for raising this critical point. Indeed, we found that IL-33 decreased at the mRNA level in our asthma samples. The regulation of IL-33 expression is very complex and beyond the scope of this study. We confirmed the decrease of IL-33 in our separate cohort using Nanostring GeoMX platform (new Fig. 2g). In addition, we performed analysis of publicly available single cell RNA-seq data set (*Vieira Braga et al. Nature Medicine 2019*), and we showed that IL-33 was again decreased in all cluster expect a subset of basal cells (Fig. 3). Given that IL33 has a complex and nontraditional secretion pathway, it may well appear that IL-33 transcript is indeed reduced in asthma. However, due to the altered spatial organisation, we think that the local concentration of IL-33 is likely very high in specific areas and will likely drive enhance immune activation. We have added these points in our discussion.

Fig. 3: Expression of IL-33 in single cell RNA-seq. Analysis of publicly available dataset (*Vieira Braga et al. Nature Medicine 2019*). IL-33 expression in epithelial cluster (a) and non-epithelial clusters (b)

12. L120-123: It is essential to detail the specific medications used by the patients, including their dosages and duration of treatment, to establish a correlation with the observed data. Analyzing these relationships can provide insights into how treatment protocols contribute to the changes in biomarkers observed in the study. Please explain it.

13. Are all asthma patients (endotypes) from whom biopsy samples were taken homogeneously, and was statistical analysis conducted considering factors such as drug resistance, race, gender, age, and specific exposure? Please explain it.

These are all important comments, we have now provided a simplified table with the critical clinical information (Supplemental Table 1 and 2). All samples were taken homogeneously by the same operator per cohort, and we did not detect batch effect between samples. This initial study reports the fundamental mechanisms at the lung wall during health and asthma that change the tissue organisation. As such it was not powered to detect effects of ethnicity, gender, exposure or age. Further studies will need to determine the importance of these factors, but we believe this is beyond the scope of the current study.

14. How was the impact of receptor-ligand interactions in spatial interactions between cells, as well as endocrine and paracrine relationships, evaluated in the study?

We conducted spatial interaction analysis and showed how specific cell clusters producing chemokines are clustering together. We have now extended this spatial relationship for cells from a distance of 15 μm (i.e. paracrine) to 150 μm (endocrine). Our spatial drug to cell analysis (Fig. 7) provided a receptor ligand interaction and showed which drugs would be likely to interact in which part of the tissue.

15. L139-141. Please compare with non-asthmatic samples and explain 'within close distance' quantitatively.

This part of the result has been modified to include a broader distance analysis as suggested by the reviewer. Please also see comment 3 of the reviewer #1.

16. L155-157. "Mast cells were abundant within the lung biopsies" Please re-write it and check.

In addition to modifying the text as suggested by the reviewer, we have now also significantly improved this part of the manuscript by providing additional data from an immunofluorescence study showing the enrichment of mast cell in normal lung tissue. We believe that the abundance of mast cells in the human lung is underestimated, likely due to digestion methods, and we think that they likely play a critical role in controlling lung immune response during homeostasis. Our future studies will investigate this further.

17. L166. "VEGFR2 and IL-7R that are vital for IL-33, vascular" Please clarify if this is the finding of the current manuscript.

The expression of these two receptors by mast cell was intriguing however we do not currently have a hypothesis as to why mast cells would express them. This observation has been removed for the new version of the manuscript and will be followed up in further studies.

18. L184. (our annotated mucous gland areas) The exact location from which the biopsy samples are taken is important because there are differences in cell types and mucus density in the sections of the lung parenchyma and bronchi. Please discuss it.

All our biopsies were taken during clinically approved bronchoscopy – they are endobronchial biopsies taken from the upper right lobe and thus represent the bronchial wall.

19. L240. The authors did not above provide information about tryptase in the upper sections of the text, but the authors did not conduct any measurements, or they did not mention or emphasize this in the manuscript.

We apologised if this information was not clear. We did not evaluate tryptase level in the current study as it was performed in the original clinical trial published in the New England Journal of Medicine (*Cahill et al. NEJM 2017*). They observed that serum tryptase was decreased by $42.7 \pm 31.6\%$ in the imatinib group whereas the placebo group showed a $11.5 \pm 31.0\%$ decrease (p value = 0.02). We have now better highlighted this in the figure and manuscript in the revised text.

Reviewer #3 (Remarks to the Author):

The authors use spatial transcriptomics technology to examine gene expression in situ in endobronchial biopsy sections from patients with asthma and controls. The application of this technology in this way is powerful, but the paper reads more like a feasibility result for the utility of the technology rather than one that uncovers new findings. In particular, the abstract text is overly committed to emphasizing the utility of the spatial transcriptomic technology at the expense of providing more granular detail about the main study findings. Other concerns include insufficient detail about the human subjects numbers, the collection and quality assurance of biospecimen collection and quality control, and the rigor of statistical analysis.

We appreciate the comments of the reviewer and agree that this new technology is powerful to decipher original mechanisms of lung biology, but we apologies for the phrasing of the study in the abstract. The new version of the manuscript has been improved to focus on the message of the paper and highlight our scientific discoveries. The main finding of our study is the existence of discrete areas within the lungs enriched in cells producing alarmins and chemokines. These cellular ecosystems exist in healthy individuals potentially for protection against pathogens and are dysregulated during asthma. We characterized which cells are contained within these hubs, which mediators are produced, and highlight potential mechanisms as to how they are formed (i.e. mast cells and AREG) and finally detail how drugs interfering with mast cell activity could facilitate tissue regeneration.

1. Human subjects.

- the supplemental table 1 is a densely populated excel spreadsheet, not the summary table that readers need. A table that summarizes the clinical and demographic features of the asthma patients and controls should be provided.

- the main manuscript needs to indicate how many asthma subjects and controls were included in the study.

These elements were not clearly identified in the original version of the manuscript and are now better indicated throughout the paper in figure legends. As suggested, we have now included a simplified table with clinical and demographic data (Supplemental table 1). In addition, we have performed additional experiments with another cohort of patients and have now analysed a total of 8 healthy and 20 asthma subjects.

2. Biospecimens;

- there is little data on the biospecimens and how they were collected. How were the biospecimens obtained? How many biopsies were collected per subjects? Were any quality measures used to classify biopsies as acceptable or not?

Samples were collected during clinically approved bronchoscopy at the Royal Brompton hospital, Leicester Hospital and Brigham and Women's Hospital. Multiple biopsies were taken from each donor and only the better-preserved samples were used for single cell spatial transcriptomics (see point below). Together, the authors have decades of experience analysing if these tissues represent lung bronchial wall and all agreed that our samples achieved a sufficiently high-quality standard. Quality control measures include H&E staining to investigate cellular architecture of the epithelium, presence of immune cells and vasculature and structure of the smooth muscle layer.

- endobronchial biopsies frequently have epithelial cell denudation because of processing artefact. How was this issue handled?

The reviewer raises an excellent point we have experienced and have been careful in our analysis. All our samples were thoroughly examined for any signs of this artefact, and we did not observe major denudation. Following processing for Xenium or GeoMX, we further examined our samples and carefully avoided regions with potential damage as shown in Extended data Fig. 2 and 8. This selection was essential for us to draw meaningful conclusions related to spatial organisation of cells.

- submucosal gland are variably present in endobronchial biopsies – was any methods feature included to determine when glands could be analyzed?

Indeed, the presence of submucosal gland was not observed in all our samples and we did not find a high proportion of them in the samples used for Nanostring GeoMX. To determine if glands were present, we employed both the H&E staining showing clear gland morphology in section as shown in Fig. 2a and Fig. 4a, the presence of specialised cell populations such as serous cells enriched in these structures and finally the spatial expression of mucous related genes such as MUC5B.

3. Imatinib trial biospecimens and details

- details are needed for how many human subjects were included and how many days or weeks were patients treated for. What drug dose was used?

The original trial included 62 patients (30 placebo and 32 imatinib), and they were all treated for 6 months with 400 mg of imatinib or placebo once daily. As this trial and samples were collected 10 years ago, we obtained workable samples from 8 patients (4 placebo and 4 imatinib) at the end of the trial. One sample (imatinib group) was however lost during the processing of the Xenium slide.

We have now better highlighted this information in the revised version of the manuscript.

4. Mast cells data

- the number of mast cells is very low on the UMAP plots (Figure 2 and Figure 3). What is the actual number of these cells? Can authors provide more information about cell numbers in total and in the different samples?

We have analysed a total of 92,963 cells within our Xenium experiments and confirmed data using an independent cohort and different platform (i.e. GeoMX). Among these cells, 4,231 were mast cells which we believe is significant number of cells analysed. Recent published single-cell RNAseq studies (from digested tissues) used comparable or considerably fewer cells to characterise these cells: 2,690 mast cells (*Tauber et al. J Exp Med 2023*), 332 lung mast cells (*Rönnberg et al. Front Immunol 223*), 7355 nasal polyps mast cells (*Dwyer et al. Sci Immunol 2021*).

Details of cell number per donor is provided below:

Donor	Pathology	Treatment	Total Cell analysed	Mast cell number
1	Healthy	NA	10153	625

2	Healthy	NA	9160	404
3	Healthy	NA	8759	608
4	Healthy	NA	2594	127
1	Asthma	NA	8028	353
2	Asthma	NA	8057	593
3	Asthma	NA	8150	442
4	Asthma	NA	5667	226
1	Asthma	Placebo	10516	96
2	Asthma	Placebo	586	18
3	Asthma	Placebo	6515	176
4	Asthma	Placebo	4342	105
1	Asthma	Imatinib	2604	74
2	Asthma	Imatinib	4813	328
3	Asthma	Imatinib	3019	56

- mast cells are not shown in the images in Figures 2 and 3, so the "clear abundance of mast cells in proximity to the most pro-inflammatory areas" is hard to evaluate and visualize.

We have now improved this aspect of the manuscript and data relating to mast cell abundance including 3D imaging of lung tissues are now present in Figure 5. We believe that mast cell abundance in human lung is underestimated in many studies, and we hope to change this paradigm.

5. Mucin gene data

- usually patients with asthma show an increase in gene expression for MUC5AC and a decrease in MUC5B. The authors find different results in their analyses for MUC5B. Was MUC5AC included in the panel? Can the authors comment in more detail on this observation and refer to the literature?

MUC5AC was not included in our Xenium panel however we performed additional analysis from our additional cohort with full transcriptome and confirmed that MUC5B was enriched in severe asthma samples and showed an intermediate level in mild asthma samples (Fig. 4f). MUC5AC showed a similar increase in mild and severe asthma however it was not significant.

Whilst an increase in MUC5AC and a decrease in MUC5B expression have been reported during asthma (*Khorasani et al. Respiratory Medicine 2023*), other reports indicate that both genes are increased (*Tajiri et al. Allergology International 2022*). Discrepancy in the literature could be due to different types of patients and samples analysed. Most studies relied on fluid analysis (e.g. BAL or sputum) that may not entirely reflect the cellular production at the lung bronchial wall. Here, our data from 3 cohorts (Healthy/asthma Xenium, Healthy/Asthma GEMX and Asthma placebo/imatinib) indicated the same trend toward increased MUC5B.

As suggested, we have now added these elements to our discussion.

6. Basal cell clusters

- how do the authors justify that the 2 basal cell clusters cannot be found in all datasets? Are they biologically relevant? Have they been described in other studies?

Our second dataset did not contain the two basal cell clusters. This could be due to the nature of the samples the trial did not generate healthy samples as it would have been unethical to treat healthy

individuals with imatinib. Consequently, we only analysed samples from patients with asthma. Imatinib had a profound impact on these cells and may have led to a strong reduction of the proportion of the pro-inflammatory cluster.

We performed additional analysis from single-cell RNAseq (*Vieira Braga et al. Nature Medicine 2019*) shown here (Fig. 4). Two distinct populations of basal cells were described, and they associated them with different level of maturity. The population of basal cell 1 was also characterized by high expression of F3 like in our dataset. Here, we made a significant further step in their characterisation by showing their spatial segregation and production of alarmins and chemokines.

Fig.4: Basal cell cluster in a publicly available dataset.

7. Chemokines and alarmins

- the decrease in chemokines and alarmins in asthma is surprising. Was the number of transcripts per cell the same between cohorts?

We strongly agree with the reviewer about this observation and this result was surprising for us as well. We did not detect differences in the number of transcripts globally between cohorts. Healthy cells have ~41.4 transcript/cell and asthma cells ~38.96 transcript/cell which does not explain the reduction in specific mediators.

8. Data presentation and interpretation

- percentages should be added to the graphs showing the distribution of cellular neighborhoods because the increase or decrease in interaction between cell types is not always clear. There is currently very little visualization of sample distribution.

Following suggestions from all reviewers, we have significantly improved the clarity of all figures including addition of percentages and statistical significance.

- remodeling signature; that authors state that: "However, we observed strong tissue remodeling signatures with reduced distances between inflammatory cells and stromal cells" (lines 179-180). What exactly is the remodeling signature? Can authors bring more information on how they jump to this conclusion ?

We agree with the reviewer that “tissue remodelling” is not a precise term as it involves many factors, and criteria vary widely between different studies. Here, remodelling refers to the series of events leading to change in tissue function and organisation. We believe that during asthma specific combinations of cells “inflammatory ecosystems” (e.g. basal cell 1 + endothelial cell 2 at the epithelium and fibroblast + endothelial cell 2 in the mucous gland) are modified by the local chronic inflammation leading to changes in cellular architecture (e.g. cell becoming more closely located). Mast cells are central to this phenomenon as they are present in these ecosystems and express key molecules such as AREG capable of modifying tissue architecture. The revised version of the manuscript clarifies these ideas.

- the authors discuss pro-inflammatory ecosystems, but it seems that these ecosystems are only present in healthy samples, as shown by the decrease expression of chemokines and alarmins in asthma samples.

Indeed, we think that the normal homeostasis of the lungs involved the constant presence of these ecosystems to maintain low level of immune recruitment to protect the tissue from infectious agents. Our data are in line with other studies notably from the lab of Donna Farber showing specialised areas in the bronchi where B cells are functionally active. How these cells reach the tissue and know where to position themselves require stromal cells acting together to secrete chemokines as shown in our study.

During asthma, and despite strong anti-inflammatory treatments, we noticed that the structural cells continue to aggregate together and are still able to secrete proinflammatory mediators. One could imagine that stopping treatment may lead to a return of these ecosystems at their maximum capacity and lead to deleterious impact for the lungs.

- when the authors discuss increases or decreases in gene expression or spatial interaction, these differences are not justified by statistical tests and seem to refer to simple observations.

As suggested, we have now improved all the figures and incremented statistical visualisations for gene expression and distance analysis.

Point by point response to Comments on NI-A38560A-Z

The modified text in the manuscript is highlighted in red.

Reviewer #1 & 2: We would like to thank again the reviewers for their insightful comments, and we are happy that our revised version satisfies their comments.

Reviewer #3 (Remarks to the Author): The authors provide a revised manuscript with additional data from 10 more patients with mild/severe asthma and 5 healthy controls and there is improved statistical analyses and rigor. I have the following comments:

1. It is not clear that the main message of the paper advances the field. The authors state that their main finding is that the airway wall in health and in asthma has discrete cellular areas (“cellular ecosystems”) that are enriched in cells producing alarmins and chemokines. Surprisingly, these cellular ecosystems were more prominent in healthy controls than in asthma prompting the authors to speculate that they protect against pathogens and are dysregulated during asthma. But this is not convincing because their data show that the alarmins that are thought to drive type 2 inflammation in particular are decreased in asthma biopsies. It is possible that there is a negative feedback or some other explanation for this finding but the lack of clarity is a weakness.

The reviewer is correct in summarising our study. We provide the first evidence of these cellular ecosystems in human samples in health and asthma using a new technology with enormous potential. The decrease in alarmin and chemokine production is surprising, and we discussed extensively this observation as suggested by the reviewer in the previous version of the manuscript (line 434 to 444). We apologise if this section was not clear enough and we have now modified our discussion to provide better clarification (line 431-440) including the possibility of negative feedback loop. We strongly believe that our study is focussed on the very specialised areas of alarmin and chemokine production and our results demonstrate that cells producing these mediators cluster together, thus leading to increased availability of pro-inflammatory mediators locally within the specific niche.

2. The data for cell characterization within the ecosystems or hubs is presented in a confusing manner and the figures, legends and manuscript text remain hard to read through and fully understand. There is a strong reliance in the paper on general statements about spatial mapping and landscape description for its own sake.

Due to its novelty, we agree with the reviewer that data visualisation for single cell spatial transcriptomics data is challenging, and we have continually aimed to improve our representations and graphical visualisation. No standard formats are acknowledged by the scientific community but we employed representations widely accepted in the field of lung spatial analyses, for example with protein staining (*Sariyar et al. Nature Communications 2024* <https://doi.org/10.1038/s41467-024-53752-x>). Furthermore, as all data are publicly available, we look forward to see how the scientific community will explore our datasets in the future.

As confirmed by Reviewer 1 and Reviewer 2, all our data are supported by strong statistical analyses and a robust number of patients. As previously mentioned regarding human studies, it is complex to derive strong mechanistic insights from human samples, and we have interpreted our observations in the light of our extensive knowledge of asthma and type 2 immune responses in the lungs.

3. Although additional human characterization data is provided in Supplementary tables 1 and 2, the information provided is hard to assimilate because data about inhaled medications generally missing and the lung function data is inconsistently presented (FEV1 data for some and FEV1/FVC ratio for

others. It is not clear how patients were subdivided into mild vs severe asthmatics, a subgrouping which is used in some of the figures.

We apologise if the clinical information provided was not clear enough. We have revised Supplementary Tables 1 and 2 and all figure legends to clarify that patients with asthma are receiving inhaled corticosteroids in accordance with standard clinical guidelines and from which cohort of patients the data are presented (i.e. cohort #1, #2 and #3). For patient stratification between mild and severe asthma, severe asthma (n=4) was defined by the European Respiratory Society (ERS) and American Thoracic Society (ATS) consensus criteria and all participants mapped to GINA 5 (track 2) of the current GINA 2024 guidelines. Patients with mild asthma were GINA steps 2-3 on either low dose ICS (1/3) or low dose ICS/LABA (2/3).